# Possible Synergies of Nanomaterial-Assisted Tissue Regeneration in Plasma Medicine: Mechanisms and Safety Concerns

**DOI:** 10.3390/nano12193397

**Published:** 2022-09-28

**Authors:** Priyanka Shaw, Patrick Vanraes, Naresh Kumar, Annemie Bogaerts

**Affiliations:** 1Research Group PLASMANT, Department of Chemistry, University of Antwerp, 2610 Antwerp, Belgium; 2Department of Medical Devices, National Institute of Pharmaceutical Education and Research, Guwahati 781125, Assam, India

**Keywords:** plasma medicine, reactive oxygen species, reactive nitrogen species, pulsed electric field, endogenous electric fields, nanotoxicity

## Abstract

Cold atmospheric plasma and nanomedicine originally emerged as individual domains, but are increasingly applied in combination with each other. Most research is performed in the context of cancer treatment, with only little focus yet on the possible synergies. Many questions remain on the potential of this promising hybrid technology, particularly regarding regenerative medicine and tissue engineering. In this perspective article, we therefore start from the fundamental mechanisms in the individual technologies, in order to envision possible synergies for wound healing and tissue recovery, as well as research strategies to discover and optimize them. Among these strategies, we demonstrate how cold plasmas and nanomaterials can enhance each other’s strengths and overcome each other’s limitations. The parallels with cancer research, biotechnology and plasma surface modification further serve as inspiration for the envisioned synergies in tissue regeneration. The discovery and optimization of synergies may also be realized based on a profound understanding of the underlying redox- and field-related biological processes. Finally, we emphasize the toxicity concerns in plasma and nanomedicine, which may be partly remediated by their combination, but also partly amplified. A widespread use of standardized protocols and materials is therefore strongly recommended, to ensure both a fast and safe clinical implementation.

## 1. Introduction

In the past decennium, plasma medicine has presented itself as a young, yet promising application of plasma technology for biomedical purposes [1,2]. The domain is often distinguished into two parts, based on whether the plasma is used indirectly, e.g., for prosthesis surface conditioning [3,4,5] or nanomaterial synthesis [6,7], or is directly in contact with the biological tissue to be treated [8,9]. Alternatively, it can be subdivided according to the field of application, such as sterilization [10,11], dermatology [11,12,13], wound healing [11,14,15], dentistry [11,16,17] and oncology [11,15,18,19]. As this list illustrates, plasmas form versatile tools for the modification or stimulation of both inanimate and biological matter. Not surprisingly, they can even change the bulk condition of liquids or soft condensed matter through the transfer of plasma-produced reactive oxygen and nitrogen species (RONS), radiation, and electromagnetic fields. These stimuli, therefore, do not only affect the biological tissue under treatment, but potentially also any foreign material in it or attached to it, such as nanoparticles or a nanostructured prosthesis contact surface. This brings forward a first motivation to study the possible synergistic mechanisms of nanomaterials in plasma medicine, in terms of opportunities, as well as safety concerns.

In comparison to plasmas, nanomaterials have a much longer and more established history in biomedicine, to a great extent for very similar applications. Their versatility is readily explained with the various functions they can perform, such as drug delivery [20,21,22], bioimaging [23,24], electrochemical sensing [25,26], oxidative stress regulation [27,28,29], nanocatalysis [30,31], and local electric or magnetic field modulation [32,33]. In principle, each of these functions can be employed to improve the aforementioned plasma therapies, for instance by fine-tuning the redox chemistry or the electric or magnetic field interaction. Conversely, plasma treatment allows to modify the nanomaterial surface and environment, in order to regulate the properties of the foreign material in situ. This calls forth a second motivation to scrutinize potential synergies between both technologies.

Several review papers have already been written on the expected or observed synergistic combination of nanomaterials and cold atmospheric plasma (CAP) therapy. In 2011, Kong, Keidar, and Ostrikov provided an early and still recommended general perspective on possible synergies in terms of reaction chemistry, cell permeation, and cellular manipulation, as well as enhanced or selective tissue penetration [34]. A review by Keidar and collaborators and a topical review and editorial by Keidar alone followed in 2015 and 2016, each with a short update on observed anti-cancer synergies of plasma and nanoparticles (NPs) in experimental research [35,36,37]. In 2017, Aryal and Bisht devoted a mini-review on the synergy between gold-NPs and CAP for cancer treatment, suggesting oxidative stress as a possible origin [38]. One year later, Smolková et al. compared the advances, limitations, and misconceptions regarding magnetic NPs, plasma, and lasers as individual physics-based biomedical technologies [39]. Although they did not discuss the combination of these techniques, their conclusions inarguably apply to it, indicating some of the major challenges for future research. They namely pointed out the wide variety of both the configurational and operating parameters and the treated biological test systems as a paramount obstacle for comparing results from different investigations and identifying the underlying mechanisms. They, therefore, advocate a more coordinated approach with standardized protocols and materials, in order to ensure a faster scientific progress and industrial adoption. Clearly, this becomes even more important when combining two biomedical technologies as versatile as CAP and NPs.

More recently, a few reviews added further insight into the synergistic combination of plasma and nanomedicine. Kaushik et al. reported recent updates for several biological applications, including plasma synthesis of biologically relevant nanomaterials [6]. In a topical review, Ostrikov and collaborators selected gold-, graphene-, and liposome-based NPs as representative nanomaterials for oncotherapy and briefly debated possible synergies with CAP [7]. An expert opinion was given by Liu, Szili, and Ostrikov, to establish a nexus between plasma, nano- and digital technologies in healthcare [40]. They briefly discussed the synergy between the plasma- and nano-segments of the triangular relationship on a general level in terms of disease diagnostics, drug delivery, multimodal treatment, photodynamic therapy, acidity regulation, thermal effects and magnetic field influences. Rasouli et al. emphasized the complementary features of plasma and nanotechnology for cancer treatment, as well as the need for standardization of procedures to reach faster clinical implementation [41], in clear similarity to the aforementioned message of Smolková et al.

While these reviews provide a nice overview of the state-of-the-art, their focus remains either on the general picture or anti-cancer effects. Less attention has been paid up to now to the synergy of plasmas and nanomaterials in tissue engineering and regenerative medicine. Nonetheless, this subdomain may be considered an overarching discipline with respect to most—if not all—of the envisioned biomedical applications, since it closely relates to sterilization and plays a pivotal role in dermatology, wound healing, dentistry and oncology. Note in this regard that any regenerative therapy with anti-cancer effects forms an attractive post-treatment method in oncology, e.g., after surgical removal of a tumor. Since both plasma and nanotherapy fall into this category for a broad range of biological materials, their combination is especially promising as such a post-treatment technique in oncology. As should be emphasized, their regenerative and anti-cancer properties extend over a broad spectrum of biological materials, ranging from skin [12,42] to bone [43,44,45] and even neural tissue [46,47,48,49]. This presents a third motivation to investigate their possible synergies, specifically for tissue engineering and regenerative medicine.

Combining plasma and nanotechnology for tissue regeneration has already been suggested in the literature on a few occasions. Ostrikov’s team, for instance, published a review on stem cell control by means of nanotechnology, presenting plasma as a versatile tool for the synthesis and processing of nanomaterials, as well as for direct treatment of stem cells [50]. Next to that, Zarrintaj et al. proposed to integrate plasmas and other physical stimuli with nanostructured wound dressings in regenerative medicine [51]. Emmert et al. reviewed CAP for wound healing and similarly suggested a combination with biocompatible and biodegradable nanofibrous scaffolds [52]. However, the possible synergies between the two technologies specifically in tissue engineering and regenerative medicine are not commonly mentioned or even alluded to. Moreover, an overview of these possibilities, as well as a research strategy to verify them, is still missing. This forms a fourth motivation to explore the similarities and complementary features of nanomaterials and plasmas for tissue regeneration in a well-structured manner.

With the present perspective article, we aim to advance a concise fundamental basis for this application domain, in order to propose a few directions for future research. Although it does not lie in our ambition to be fully comprehensive, we intend to illustrate how the nanomaterials may be matched to the plasma source, or vice versa, based on their corresponding individual physicochemical features and induced biophysical and biochemical processes, to obtain a desired effect. This namely appears the most appropriate approach at present, considering the very scarce experimental research for now on the simultaneous use of plasmas and nanomaterials for tissue engineering and regenerative medicine. Where applicable, we will discuss the few empirical studies on this topic that we could find in the literature. Our exploration will focus in particular on the influence of the plasma-produced redox chemistry and electromagnetic fields on the nanomaterial behavior, without intending to be exclusive. In this way, we hope to invoke a higher awareness and appreciation for the opportunities in the overlapping area of nano-, plasma and regenerative medicine, in order to stimulate more multidisciplinary research on these challenges.

## 2. Nanomaterials in Regenerative Medicine and Tissue Engineering

Regenerating tissues and organs after injury to induce complete structural and functional repair is clinically challenging, because human organ systems display some of the poorest regenerative abilities [53]. To overcome this problem, the regeneration process can be assisted by providing a configurational matrix or mechanical support to the biological material and regulating the local cellular stress and biotic processes. Nanotechnology permits each of these functions to be performed by a single substance. The second advantage of NPs in tissue regeneration relates to their small size, which is comparable to the small biomolecules such as extracellular matrix components that further allow cell migration and re-epithelialization near the wound sites [54]. They can, therefore, quickly penetrate through biomembranes and alleviate cellular uptake [54,55]. 

Thirdly, various NP composite scaffolds have been shown to stimulate cellular functions by means of their electrical conductivity and mechanical stress, which is beneficial for tissue regeneration [56,57,58,59]. It is possible to give an interesting feature to NPs, such as fluorescence or electromechanical properties, and hence, to make them very useful in tissue regeneration applications. Even magnetic NPs have been helpful for tissue structure. Fourthly, NPs with an appropriate size and composition can activate cells through oxidative stress-mediated pathways [60,61,62]. NPs can namely induce oxidative stress by entering the cell and generating intracellular reactive oxygen species (ROS) through various mechanisms. As further explained in Section 4.1 and Section 4.5, these two abilities depend on several properties of the NPs and cells. Many studies have linked an overproduction of ROS to various chronic and degenerative diseases, including cancer, insulin resistance, diabetes mellitus, atherosclerosis, aging, respiratory, neurodegenerative, and digestive illnesses [63,64]. At moderate levels, however, they can promote tissue regeneration. This dual role of oxidative stress relies on the amount of ROS [2,65]. Increased intracellular levels cause severe biological dysfunction, and even cytotoxicity [66]. In contrast, low ROS concentrations implicate cell survival, proliferation, differentiation, and regeneration [2]. Healthy cells actually employ this effect to activate their regeneration capability. Usually, they achieve this by producing intracellular ROS through the mitochondrial electron transport chain reaction and nicotinamide adenine dinucleotide phosphate (NADPH) oxidase, regulated by the antioxidant glutathione machinery [67,68]. In this way, the ROS and antioxidant chemistry play a crucial role in many biochemical pathways, cellular biosynthesis, and regulation processes during tissue regeneration at the molecular level [65]. These dual properties of ROS allow to develop novel therapeutic approaches in regenerative medicine and tissue engineering [2]. According to recent studies, ROS also acts as a secondary messenger in cell signal transduction activated by cellular stress [2]. 

With regard to the oxidative stress-mediated signaling, gold-NPs and titanium dioxide NPs in particular display superior biocompatibility and surface properties to induce cell stimulation and regeneration. One study showed that gold-NPs with diameters of 30–50 nm induced human adipose-derived stem cell differentiation [69]. Another study demonstrated 20 and 40 nm-sized gold-NPs to activate the osteoblastic function [70]. Numerous other investigations have brought forward promising in vitro results using gold-NPs and titanium dioxide [71]. When carried out in animal models, however, these experimental results could not be reproduced, and hence, it has prompted researchers to search for promising alternative substitutes. 

To mitigate these problems, conductive hybrid scaffolds with implanted NPs were applied to in vivo models, in order to initiate tissue differentiation. For example, when gold-NPs and titanium dioxide were placed within a scaffold such as gelatin and hydrogel, a significant impact was observed on stem cell differentiation, cardiomyogenic differentiation, and myocardium regeneration [71,72,73]. For mechanical stimulation of the cells, NP-embedded nanocomposite polymers demonstrated superior mechanical properties for tissue regeneration relative to scaffolds without NPs [74]. Enhanced regenerative effects have been obtained by hydroxyapatite-NPs mixed with silk fibroin, magnetic NPs embedded in the gelatin scaffolds and hydrogel microfibers encapsulating poly(*N*-isopropylacrylamide) and polyvinylpyrrolidone-coated titanium dioxide NPs [72]. Similarly, carbon nanotubes and graphene-based NPs conjugated with polylactic acid and polylactic acid combinations have also been used to induce human periodontal ligament mesenchymal stem cells differentiation [75]. 

Nanocomposite scaffolds with sophisticated electrical and mechanical features can thus be used to balance the redox and biochemistry. The enhanced electrical and mechanical stress induce intracellular ROS [76]. This influences the pathways of mitogen-activated protein kinase—extracellular-signal-regulated kinase (MAPK-ERK), signal-regulating kinase-1 (ASK1), thioredoxin (Trx), activation of receptor tyrosine kinases (RTKs), and signal transducer and activator of transcription (STAT), which all play a vital role in tissue regeneration [77,78,79]. In compliance with increasing evidence, MAPK regulates the cytoskeletal organization, cell shape changes, cell death, and proliferation through ROS-mediated oxidative stress [80]. Under normal conditions, for instance, MAPK inactivates the ASK1, which tends to form a complex with Trx. Under conditions of elevated oxidative stress, however, the sulfhydryl groups present on Trx become oxidized, ASK1 is released, and p38 and c-Jun N-terminal kinase (JNK) activity is induced [79,81]. 

Increased electrical and mechanical stress can also lead to cytokine production and the expression of growth factors [82,83]. In turn, platelet-derived growth factor induces intracellular generation of ROS, which is believed to be a critical factor in cell proliferation [84,85]. These growth factors also trigger the kinase family MAPK-ERK via a ROS-dependent mechanism [86,87,88]. Moreover, the platelet-derived growth factor is significant for atherogenesis by activating different proatherogenic smooth muscle genes [86]. 

In addition to the aforementioned signaling pathways, a conducting nanocomposite scaffold modifies the endogenous electric fields, leading to charge redistributions and thus a secondary type of electrical stress. As a particularly interesting effect, this will induce a transmembrane ion flow through the activation of ion gated channels and transporters [89]. These channels are specialized in the transport of specific ions, such as Na^+^, Cl^−^, K^+^ and Ca^2+^. They reside in all cell membranes and are essential for membrane polarization and cell response during stress conditions [89,90]. They are especially sensitive to electrical stress, which therefore triggers such transmembrane ion flow, resulting in cytoskeleton changes and directional cell migration. In the outcome, the ion influx contributes to persistent cell regeneration [91]. Other possible effects of exogenous electric fields will be discussed in detail in Section 3.3.

Despite the promising potential of NPs in regenerative medicine and tissue engineering, still, it remains challenging to adopt them into clinical practice. First and foremost, the biological repercussions of NPs are strongly dose- and exposure-dependent. Beyond a critical amount, undesired effects lead to toxicity, carcinogenicity, and teratogenicity. Even though NPs are administered below their threshold concentrations, and they are considered acutely non-toxic, prolonged exposure may result in chronic ramifications. Additionally, bioaccumulation inside the body enables the NP concentration to reach elevated levels toxic to cells, promoting cancers, or deleterious effects on other body parts, including foetuses before their birth [92]. Thus, to reduce the therapeutic dose and associated unspecific toxicity of NPs, the combination of nanotechnology with CAP provides a novel strategy to overcome the current limitations of NPs.

## 3. Plasma Medicine for Regenerative Medicine and Tissue Engineering

### 3.1. The Basic Working Principle behind Biomedical Plasma Treatment

Plasma is a partially ionized gas composed of free electrons, (positively and negatively charged) ions, and neutral particles. Its temperature is determined by the thermal motions of the free electrons and the heavy particles. However, often these temperatures cannot be expressed as one value. In the case of high densities, all heavy species (such as atoms and ions) will approach thermal equilibrium due to their frequent collisions, giving rise to a common gas temperature. In many applications, these plasmas occur under atmospheric pressure for the ease of use. In a low current regime, as generally used for biomedical applications, the thermal equilibrium is not maintained between free electrons and heavy particles. The temperature of the heavy particles (such as atoms and ions) is accordingly much lower than the temperature of the electrons, as the latter are more easily accelerated by the external electric field due to their much lower inertia. This class of plasmas is called cold atmospheric plasma (CAP). The biomedical application of CAPs often requires the heavy species to remain close to room temperature, a condition which is easily met for dielectric barrier discharges and plasma jets [93,94,95]. These plasma sources can interact with biological materials without causing thermal or electric damage to the tissue [96], while supplying its surface with a broad spectrum of reactive species. 

The electron energy distribution in non-equilibrium discharges forms the engine behind the plasma chemistry. The excited species and radicals are produced due to this electron impact, while electrons are multiplied in an avalanche, as a result of which there is an increase in the generation of reactive gaseous species. Simplified, the important components of CAP are summarized in the following steps [97]: An external electric field accelerates the electrons, resulting in ionization, excitation and dissociation of the feed gas (usually an inert gas such as argon, helium, or a mixture of different gases such as ambient air, oxygen, nitrogen) via electron impact, resulting in ions, excited species, and radicals or smaller molecules/atoms, respectively.During the plasma discharge, highly charged atoms, molecules, and free electrons subsequently interact with other particles (ambient air, liquids, surfaces), and they can generate secondary and tertiary reactive species.The corresponding excitation and depletion processes and the charge transport eventually lead to the formation of UV, visible light, and IR, which gives the plasma its characteristic color.

Although it is not possible to “store” plasma in a gaseous state, plasma can be maintained as long as the energy supply exists. As a common attribute of all plasma sources developed for biomedical applications, the major reactive molecules produced in CAPs emerge when the components of the partially ionized gas (atoms, molecules, radicals, ions, and electrons) interact with the molecules of the surrounding air, such as oxygen (O_2_), nitrogen (N_2_)_,_ and water vapor (H_2_O), and with the biological sample. This sample generally has a wet surface, not only for cells in a medium, but also in the case of biological tissues, because they are naturally surrounded by a liquid layer. As such, the reactive species are further transferred into the medium through diffusion, drift, and liquid convection (see also Section 3.2). Similarly, the plasma–liquid interaction induces electrical coupling between the gaseous and condensed phases, as further discussed in Section 3.3. The plasma composition and the subsequent effects on cells can thus vary enormously depending on the plasma source, the plasma settings, the ambient conditions and the biological target [98].

In general, there are two ways to administer CAP treatments in biomedicine (Figure 1). In the first method, known as direct CAP treatment (Figure 1a), the plasma treats the substrate directly. All the CAP components, such as electrons, atoms, molecules, radicals, excited species and ions, as well as (UV, visible, IR) radiation, and electromagnetic fields, will have the chance to affect the biological material, depending on their dose, penetration depth and lifetime. In the case of in vitro direct CAP treatment, the cells are covered with either cell culture media or other biological solutions (e.g., deionized water [68], buffer solution [19,99], or ringer’s lactate [100]). In vivo application, on the contrary, is usually performed without externally supplied liquids, therefore making the reactive species travel through the naturally present fluids, as in wound healing.

Alternatively, reactive species can be supplied to the sample in the form of plasma-treated liquid, i.e., in the absence of direct contact between the sample and a plasma. This forms the second group of methods, altogether referred to as indirect CAP treatment. In analogy with the direct variant, the implementation can take place in vitro or in vivo. This permits several other administration routes that are inaccessible by the direct method, including intravenous or hypodermal injection, topical application, and as an aerosol or spray for its use in the airways. Nonetheless, plasma-treated liquid only contains long-lived reactive species, i.e., dissolved ions and molecules, whereas radicals and other metastables have recombined before its employment.

Therefore, the core area of plasma medicine is to apply CAP in the direct treatment of biological tissues and living cells. For such biomedical applications, we can distinguish three different types of CAP sources (not to be confused with the administration techniques): direct, indirect, and hybrid ones. In a direct CAP source, the living tissue serves as a grounded electrode or one on a floating potential. Biological tissue, serving as one of the active electrodes, thus participates in the plasma discharge. Thereby, the direct mode of plasma treatment is found to have a stronger effect than the indirect mode [99]. A common example is the dielectric barrier discharge (DBD) (Figure 2, right), where a dielectric barrier covers the external electrode to prevent hazardous high electrical current densities in the plasma and at the tissue surface. An indirect CAP source, on the other hand, makes use of a gas flow to transport the plasma species from their place of origin, confined between an electrode system, into the ambient air. Plasma jets (Figure 2, left) are typical examples of this methodology, used to deliver the reactants to a target that is located quite far from the point of plasma discharge. In hybrid CAP sources, a current-free condition is created in the tissue, which allows to combine the approach in the direct and the indirect CAP source [101]. 

The focus of plasma medicine is to explicitly observe the interaction between individual components of plasma such as reactive species, electric field, etc., and specific cell lines or cell constituents, to control and, ideally, normalize the therapeutic effects. Most of the plasma medicine research has been focused on cancer treatment, wound healing, and sterilization. Among the biological effects of CAP, some of the most important in the context of plasma medicine are [97]: the complete inactivation of multidrug-resistant microbes;stimulation of tissue regeneration by means of cell proliferation and angiogenesis at a lower dose of CAP treatment; andtriggering programmed cell death, primarily in cancer cells, often at higher CAP intensity and treatment time.

Cancer research encapsulates the largest subdomain of plasma medicine by far [19,35,37,66,67,93,102,103,104,105,106,107,108,109,110,111,112,113,114]. Next to that, various experimental trials have demonstrated CAP to stimulate wound healing [15,115,116,117]. Both applications can partly be attributed to its antibacterial [18,95] and antiviral effects [118,119], but generally differ in the intensity and length of the treatment (Figure 3). The underlying biological mechanisms form the topic of Section 3.2, Section 3.3 and Section 3.4.

### 3.2. Effect of the Plasma-Induced Redox Chemistry

CAP in the ambient atmosphere generates ROS, as well as reactive nitrogen species (RNS) [2,18,19,66,99,103], collectively labelled as reactive oxygen and nitrogen species (RONS). Next to their transfer from the plasma gas into the condensed material, these species can also be generated directly in the liquid phase by plasma-liquid interaction. In this way, for instance, plasma treatment of an initially unreactive solution yields a plasma-treated medium [2,18,66,68,103], often also referred to in a more general way as a plasma-treated liquid [2,99]. The most important aqueous reactive species for biomedical applications, especially with regard to cancer treatment, are superoxide anions (O_2_•^−^), hydrogen peroxide (H_2_O_2_), hydroxyl radicals (•OH), singlet oxygen (^1^O_2_), nitric oxide (•NO), peroxynitrite (ONOO^−^), nitrite (NO_2_^−^), and nitrate (NO_3_^−^). In addition, plasma-treated medium often also contains hydroperoxyl radicals (HO_2_•), nitrogen dioxide (•NO_2_), hypochlorite (OCl^−^), and dichloride anion radicals (Cl_2_•^−^) [2,18,19,68].

Plasma medicine has accordingly emerged as a new subfield of applied redox biology where biochemistry and clinical medicine are combined with plasma physics and plasma chemistry. The antibacterial and antiviral effects of CAP are, for instance, mainly attributed to the presence of RONS, leading to oxidative-related damage and modification of the phospholipid biomembrane, proteins, and DNA. Therefore, CAP might induce wound healing due to the synergistic effect of antiseptics on the wound surface in combination with tissue regeneration stimulation. According to the majority of investigations, however, the latter effect seems to be dominant. 

As frequently observed in experiments, plasma can either help in tissue stimulation or in cell death, mainly due to its exposure conditions, specifically the amount of RONS [2,108]. Accordingly, CAP might be considered as a powerful tool for specific research in redox biology as well. For this purpose, however, the redox processes will need to be isolated from other plasma-induced stimuli, such as the electric and magnetic field effects discussed in Section 3.3 and Section 3.4. The most straightforward way to achieve this is by indirect CAP treatment.

An important question posed by von Woedtke et al. in this context is why at the same time CAP works as a destructive tool for microbes, while promoting tissue regeneration in mammalian cells [97]. This difference may be due to the selectivity of the diverse reactive species to distinct organisms, in line with several experimental studies [2]. The RONS-composition generated in CAP namely turns out to be more toxic for microorganisms because of the combined effect of ROS and RNS components, whereas in mammalian cells mostly ROS are considered to be the toxic components [2,114,120,121]. Interestingly, immune cells produce RONS to eliminate microorganisms [122]. This might stimulate the neighboring tissue to generate new biological matter to replace the tissue damaged by the antimicrobial immune cell response. 

In particular, catalase and peroxidase affect ROS-signaling [123,124], showing that H_2_O_2_ is a quantitatively essential signaling species. Both H_2_O_2_ and O_2_•^−^ are implicated in activating signaling pathways [2,125]. As previously discovered in particular, low levels of H_2_O_2_ stimulate cell growth [2], cancer cells produce increased H_2_O_2_ [126] and the family of nicotinamide adenine dinucleotide phosphate oxidase is associated with cell proliferation [127], which further confirms that redox-related pathways have an essential function in cell growth control.

Intracellular ROS levels regulate redox-signaling keepers [128], such as the extracellular signal-regulated kinase, mitogen-activated protein kinase pathway, which is vital for cell proliferation [129], as well as the phosphoinositide 3-kinase signaling pathway, which is essential for cell growth and survival [2,130], and transcription factors such as hypoxia-inducible factors [131]. Indeed, many groups have demonstrated the ability of exogenous oxidants to activate the MAPK-ERK pathway [2,129]. The MAPK-ERK signal activation duration and intensity determine the cellular response outcome based on intracellular ROS contents. However, the direct correlation of ROS response to ERK activation mechanism still needs to be evaluated. Other members of the MAPK family have also been implicated as potential targets of ROS, especially to H_2_O_2_ [132,133]. More precisely, the MAPK-ERK pathway is activated by the inactivation of the phosphorylation of protein-tyrosine phosphatase 1B (PTP1B) due to the presence of intracellular ROS. It is known that PTP1B is inactivated due to the ROS-induced modification of the protein’s active pocket site of a cysteine residue [134]. In the solution case, cysteine is found to be the most reactive amino acid by plasma treatment [68,94,135].

Further, Ushio-Fukai et al. observed that reactive species act as a messenger for angiotensin-mediated signaling pathways by activating phosphatidylinositol 3-kinase [136]. In 2019, Park et al. discovered that CAP increased the proliferation of human mesoderm-derived stem cells by increasing the gene expression of various cytokines and growth factors, which further helps regeneration at wound sites [137]. Recently, we demonstrated that reactive species generated by CAP selectively initiate and amplify ROS signaling to enhance skeletal muscle cell proliferation and differentiation [138]. However, Chen et al. pointed out that excess ROS levels in wound areas can impair the antioxidant machinery and other function such as fibrosis and keratosis. As a result of that, it can cause severe inflammation in wound areas. Furthermore, the authors demonstrated that zinc-based nanosized particles significantly enhances wound healing by scavenging the Cu/Zn superoxide dismutase activity and enhancing the expression of type I collagen and transforming growth factors-β [139]. Therefore, a careful strategy must be explored for a better synergy to promote tissue regeneration.

Next to these oxidative stress-mediated signaling pathways and the ones discussed in Section 2, several other redox mechanisms can underlie the observed biological effects. Figure 4 displays two examples proposed by Canal’s group for osteosarcoma therapy [140] and by Ostrikov’s group following Jang et al. for the neural differentiation mechanism [141,142] under CAP exposure. As a third example, ROS are able to enhance drug delivery into cells by oxidizing the cell membrane [19,68,99]. Cell membrane sections rich in peroxidized lipids can namely be trafficked into the cell through membrane repairing endocytosis. This is more specifically possible with clathrin-dependent enhanced uptake useful for drug or NP delivery [143,144,145,146]. In oxidative stress, gold-NPs show higher cell cytotoxicity to glioblastoma multiforme cells via adenosine 5′-triphosphate (ATP)-dependent uptake mechanism [147]. Additionally, selective apoptosis of lung cancer via oxidative-stress mediated DNA damage using epidermal growth factor-conjugated gold particles is found to be achieved via uptake of receptor-mediated endocytosis [148]. Perhaps this also forms a vital route to transfer RONS themselves from the extracellular medium into cells. 

Gaining a more profound insight into the fundamental redox-related processes behind the biological effects will be crucial for the purpose of optimizing them further. Likely, these effects heavily rely on the relative ratios and absolute dose of the RONS in the liquid material. Based on the currently available data, we can safely assume this relationship to be strongly dependent on the cell, tissue and organism type as well. Documenting and fine-tuning the ratios and dose for various target materials will be a work of precision, complicated by the multiple possible synergies between the underlying mechanisms and the diversity in plasma sources and settings. As one of the main limitations of CAP treatment, it is nearly impossible to vary the individual RONS concentrations separately of one another. While a combination with nanomaterials will add more degrees of freedom and thus a higher complexity to the system, it may provide a feasible solution to this limitation. 

### 3.3. Effect of the Plasma-Induced Electric Fields

In addition to the chemical effects, plasma in contact with biological tissue gives rise to an electromagnetic field in the gas phase as well as in the tissue. For the sake of simplicity, we will discuss the influence of this field on the biological processes separately for the electric and magnetic components. Regarding the electric component, a first essential aspect to consider is its degree of penetration into the tissue. That is, a slowly varying external electric field will be cancelled out in the biological substance due to the rapid emergence of an electric double layer at its surface in the condensed phase. In the presence of a plasma, the situation becomes more complex with the formation of a plasma sheath in the gas phase, at the opposite side of the plasma-tissue interface. This electrical coupling mechanism between the plasma and the condensed phase makes the relationship between the voltage applied on the external electrode system and the resulting electric field distribution less straightforward. The cancelation principle for the field penetration, however, remains unchanged. The electric field generated by the charges in the sheath and, to a lesser extent, the ones in the adjacent pre-sheath and further away from the interface, is counteracted through dielectric screening by the electric double layer. The formation rate of this layer, therefore, determines the threshold frequency required for electric fields to penetrate the tissue.

Under physiological conditions, the characteristic time of this process lies around 1 µs [149,150]. In other words, a variation of the plasma sheath in the sub-microsecond scale will generate an electric field that can penetrate into the tissue. An accurate and trustworthy sheath model is therefore crucial for a profound understanding of the electric field effect. Unfortunately, such model has not been developed yet for plasma in contact with liquids or biological matter. Conventional plasma sheath models have only been constructed and tested for plasma-solid interaction. Their applicability to liquid and biological surfaces is therefore questionable, because they do not take into account several processes unique to such surfaces. Biological materials are namely always covered by a liquid layer, enabling evaporation, microdroplet nucleation or ejection, surface deformation, an ion drift-mediated liquid resistivity, an electric double layer and a distinct mechanism of electron emission. Our group recently published an elaborate perspective on how these phenomena can influence the electrical coupling, mass transport and chemistry of the plasma sheath [151]. More detailed information on the elementary processes at a plasma-liquid interface can be found in earlier reviews [107,152] and, specifically on the electron emission theory, in a recent review by Elg et al. [153].

Despite the lack of a trustworthy sheath model for a liquid or biological surface, the temporal behavior of the sheath under usual plasma treatment conditions can be deduced from experimental observations. As mentioned above, the variation of the electric field does not only depend on the applied voltage waveform, but even more on the plasma properties, which on their turn rely on the electrical coupling with the condensed phase. Sub-microsecond fluctuations of the electric field can therefore even be achieved with AC powered plasma sources having a frequency far below the megahertz range. The plume of a plasma jet namely consists of so-called plasma bullets, whereas a DBD is made up of filamentary streamers, which are both rapid phenomena. Helium plasma jets with an input power frequency around 12.65–20 kHz, for instance, were found to generate plasma bullets, transporting a field strength in the order of 1 to 10 kV/cm at a velocity up to 40 km/s when reaching a grounded electrode [154,155]. Jets powered with high voltage pulses of hundreds of nanoseconds duration at a frequency up to 10 kHz produce even faster bullets with velocities typically in the order of 100 km/s [156,157,158]. Fluctuations in the interaction of such bullets with a water surface display a duration in the order of 10 to 100 ns [158], short enough to permit electric field penetration. The streamers in DBD are even swifter, with velocities ranging from 100 to 1000 km/s and lifetimes in a solid electrode system of a few nanoseconds [159,160]. In contact with a liquid electrode, however, the plasma contact can last much longer, smoothing out the field fluctuations over the 100 ns scale [161], still well below the 1 µs threshold. Electric field penetration into biological tissue is therefore expected under usual plasma treatment conditions.

As should be noted, the rise and fall time of the electric field fluctuations largely determine the degree of field penetration, while their duration and magnitude regulate the type and size of the biological effects. On the cellular level, for instance, pulsed electric fields affect different parts of the cell according to their pulse length, as depicted in Figure 5. However, these tendencies may also partly depend on the pulse amplitude, as they were observed in pulsed electric field experiments, where a shorter pulse generally corresponds to a higher field strength [162]. The cell membrane is preferentially impacted by microsecond pulses, which last long enough to cause an intracellular ion redistribution [150]. Nuclear matter, on the other hand, forms a favorite target of nanosecond pulses, resulting in DNA and chromosome damage, altered nuclear processes and increased gene expression [163,164]. 

Other intracellular organelles selectively fall prey to picosecond pulses. Mao et al. simulated the effect of electric field pulses on a simplified cell encapsulating a smaller organelle using a finite element model, demonstrating such selectivity as a function of the pulse duration as well as its magnitude [167]. As the most experimentally studied example, picosecond pulses affect the respiratory rate and transmembrane potential of mitochondria, altering their oxidation and phosphorylation, or inducing the release of cytochrome C and activating the caspase family, which can ultimately result in apoptosis [165,166,168,169]. Changes in the endoplasmic reticulum (ER) are a second example, where an induced stress level may result in apoptosis through a variety of mechanisms [168,170]. The ER namely regulates protein synthesis, protein folding and intracellular calcium homeostasis, so abnormalities in its function can trigger the activation of certain proteins, a redox imbalance, and the release of Ca^2+^ into the intracellular medium [170]. Such elevated calcium levels enable mitochondrial depolarization, which therefore may also be a secondary effect of picosecond pulses, as sketched in Figure 6. 

Whether such effects can also be selectively induced under plasma treatment remains to be explored. Still, the influence of plasma-induced fields has already been proposed by several research groups to explain certain experimental observations. Ouf et al. and Devi et al. attributed the inactivation of fungal spores by plasmas to electroporation, i.e., the formation of reversible or irreversible pores in the cell membrane due to strong electric fields [171,172]. Tero et al. measured pores in the order of 10–1000 nm in an artificial cell membrane system after plasma treatment [173]. In an investigation on the mechanism behind plasma-induced gene transfection, Jinno et al. also indicated the electric field as a possible contributor [174]. The direct contribution of plasma-induced electroporation is, however, hard to prove, because pore formation can also result from chemical or biochemical processes. Moreover, electroporation often requires strong fields of several hundreds to thousands V/cm, as shown in Table 1. It will therefore be useful to determine down to which values this threshold field will drop in combination with these complementary effects, e.g., after oxidation of the cell membrane. In molecular dynamics simulations, lipid oxidation of the cell wall was indeed found to decrease the electric field threshold needed for pore formation [175]. 

Much weaker fields from a few to some tens of V/cm are able to induce endocytosis (see Table 1), one of the most elementary physiological functions of cells, important for the active transport of particles between the extra- and intracellular solutions. Plasma-induced endocytosis has been observed as an important mechanism in different investigations, sometimes attributed to the penetrating electric field [147,174,176,177]. Jinno et al. even found the plasma-induced electric fields to be essential for gene transfection, with a threshold value around 100 V/cm [177]. Once more, the synergy between electro-endocytosis and other effects, such as the oxidation of the cell membrane, presents an interesting topic for future research. Vijayarangan et al., for instance, found the combined effect of the plasma-induced electric field and chemical processes to be required for drug delivery into cells [176]. As a possible underlying mechanism, cell membrane sections rich in peroxidized lipids can be trafficked into the cell through membrane repairing endocytosis, with clathrin-dependent enhanced uptake useful for drug or nanoparticle delivery [144,146,147,178,179]. 

The electric field strength thus forms another decisive parameter able to selectively induce biological effects useful for the regeneration or engineering of tissues. Note in this regard that plasma treatment generally involves the deposition of a net surface charge on the biological material, both during and after the plasma contact. This surface charge is expected to generate a relatively weak, but long-living and constant, electric field in the underlying tissue, which accordingly may affect the processes therein. In line with this insight, Table 1 compares the magnitude of endogenous electric fields with externally applied fields for various processes. Natural cell and tissue homeostasis generally takes place in the presence of mesoscopic fields up to the order of 10 V/cm. Intact skin, for instance, presents a transcutaneous potential difference ranging between about 15 to 40 mV, called the skin battery, which varies according to the anatomical site [180,181,182]. The skin surface maintains a negative potential relative to the dermis, generated and sustained by active Na^+^/K^+^ ATPase pumps in the epidermis. In the case of skin injury, ions and charged particles flow out of the wound, resulting in a lateral endogenous potential difference due to the skin battery in the surrounding intact skin, as shown in Figure 7. This lateral field plays an essential role in the wound healing, and disrupting it also disrupts the healing process [183,184]. During the healing process, the lateral field declines and can therefore be used as a noninvasive indicator for the recovery stadium [182,185]. Analogous relationships are expected for burning wounds, although the healing process is likely complicated in this case by dead tissues at the injury, contamination and infection [181]. Remarkably, chronic wounds display a weaker lateral electric field than acute lesions, likely contributing to delayed healing [186,187]. Moreover, the lateral field in human skin injuries diminishes with the person’s age, explaining the higher risk for wound chronicity [185].

**Table 1 nanomaterials-12-03397-t001:** Typical values and ranges of the electric field strength naturally present in biological matter or required to induce a certain biological effect.

**Endogenous Electric Field**	**Natural Biological Phenomenon**	**Reference**
0.005–0.07 V/cm	Neuronal excitability	[188,189,190,191,192,193,194]
0.21–10 V/cm	Embryonic transneural tube potential for ion transport	[195,196,197]
0.4–2 V/cm	Laterally oriented field during wound healing	[183,198,199,200,201]
~2 V/cm	Skin battery of intact skin	[180]
**Exogenous Electric Field**	**Effect on Biological Matter**	**Reference**
0.07–0.09 V/cm	Cell reorientation	[202]
0.03–10 V/cm	Directional migration of different cell types	[182,184,203,204,205,206]
0.16–4.4 V/cm	Directional migration of neural stem or progenitor cells	[207,208,209,210,211,212]
0.1–1.5 V/cm	Directional nerve growth	[183,213,214]
0.28–2.2 V/cm	Neuron orientation in vitro	[215]
1–2.5 V/cm	Lens and corneal epithelial cell reorientation	[203,216,217]
0.17 V/cm	Keratinocyte differentiation	[218]
0.3–3 V/cm	Neuronal differentiation	[219,220,221,222]
1 V/cm	Cell proliferation in wound healing	[223]
2 V/cm	Enhanced osteoblast proliferation for bone tissue regeneration	[224]
3–4 V/cm	Depressed osteoblast proliferation	[224]
1–3 V/cm	Iontophoresis	[225,226,227]
~1–20 V/cm	Pulsed electric field for electro-endocytosis	[151,162,228,229]
400–500 V/cm	Pulsed electric field for electroporation of mammalian cells	[230,231]
~10^2^–10^4^ V/cm	Pulsed electric field for electroporation of different cell types	[151]
~10^3^ V/cm	Pulsed electric field for cell electrofusion	[162]
1 V/cm	Directional protein migration	[183]
2 V/cm	Enhanced bone protein production	[224]
≥10 V/cm	Protein crystallization	[232]
~10^4^–5 × 10^4^ V/cm	Pulsed electric field for enzyme deactivation	[233,234,235]

Since the formation of a plasma sheath at a material on a floating potential is associated with the deposition of a negative surface charge, accelerated wound healing under plasma treatment can partly be explained as the effect of the correspondingly enhanced electric field in the injured tissue. If the surface charge partially survives after the treatment with a beneficial distribution over the wound, such healing mechanism can even be long-lived. Therefore, it is crucial to assess the charge-induced biological processes, a topic that has yet received very little to no attention in plasma medicine. 

Not coincidentally, the exogenous field strength interval from around 0.01 till 10 V/cm is associated with strongly related induced effects, including the orientation, alignment, directional migration, proliferation and differentiation of cells [162]. In general, these effects strongly depend on the cell type, their environment and the characteristics of the field. Field-induced proliferation and differentiation, as an illustration, usually occur in a narrow field strength interval unique to each cell line [224,238]. Stronger fields can even counteract the natural process, instead of enhancing it [224,239]. Therefore, the electric field intervals and values presented in Table 2 should be understood as indicative only, and more exact values need to be retrieved on a case-to-case basis with dedicated experiments. Additionally, this underlines the need for standardized protocols in electric field treatment, to enable an easier comparison between studies and thus a faster clinical adoption.

External electric fields up to 10 V/cm also affect biological materials on spatial scales below and beyond the cellular level. On the tissue level, iontophoresis serves as a well-known example, used for drug delivery through the skin. Under influence of a transdermal DC voltage in the order of 0.1 to 10 V, the drug is transferred through the skin by means of enhanced passive diffusion, directional ion transport referred to as electromigration and convective solvent flow known as electroosmosis [240]. Electromigration preferably occurs through pre-existing pores, such as sweat glands or hair follicles, but can be facilitated with pore formation in the stratum corneum. Such pore formation has namely been proposed as a mechanism underlying the increased skin permeability observed after the application of transdermal voltages over 100 V or plasma treatment [151]. Electroosmosis, on the other hand, relies on the isoelectric point pI = 4.0–4.5 of skin, which lies below the physiological pH of about 7.4 and therefore favors the transport of positive ions [241]. However, this process may be disrupted under plasma treatment, as the latter decreases the local acidity. After iontophoresis experiments in vitro, an augmented skin permeability has often been observed [242]. This indicates an additional mechanism for the enhanced transport, because of structural changes in the stratum corneum. Jadoul et al. attribute this effect to four possible processes, related to increased skin hydration, localized Joule heating, local electric field enhancement, and changes in the skin integrity due to variations in the ionic constituents and pH [242].

On the molecular level, electric fields present numerous effects as well. In analogy with iontophoresis, particles may be transported under the field through biological membranes in various types of media and tissues by electrophoresis, dielectrophoresis and electroosmosis, depending on whether they are charged, polarizable or carried by the fluid drag force, respectively [243,244,245,246]. At higher field intensities, polar molecules can further be reoriented and have their conformational structure modified. Such processes are especially important for proteins, whose quaternary structure determines their function. In line with Table 1, pulsed electric fields in the order of 10 kV/cm are able to deactivate proteins by first unfolding them and next irreversibly transforming them by denaturation or aggregation [233,234,235]. At much lower intensities, the electric field-induced effects are milder in general once more, but notable on sufficiently long time scales as a redistribution or altered production of the proteins.

In a nutshell, plasma-generated electric fields may trigger and regulate a complex spectrum of biological mechanisms relevant to regenerative medicine and tissue engineering. Endogenous electric fields play an essential role in wound healing and various cellular functions essential to tissue growth, such as cell reorientation, directional migration, proliferation and differentiation. These functions may be regulated with plasma-generated electric fields and charge deposition for a wide variety of cell and tissue types, indicating the versatility of plasmas to engineer or regenerate diverse biological materials. Stem and progenitor cells serve as an attractive research material for this purpose in particular, because of their ability to self-renew and differentiate. We hereby refer to the relatively old, but still very inspirational review articles by Borghi et al. on stem cell control by means of nanomaterials or plasma treatment as separate biomedical technologies [50] and by Li and Jiang on the correlative role of stem cell niches and endogenous electric fields in tissue repair [237]. For a comprehensive overview of electric field effects in biology with a more in-depth discussion than the one we made above, we recommend the review papers by Kolosnjaj-Tabi et al. [162] and by Zhao et al. [246], as well as a recent perspective article from our group [151].

### 3.4. Effect of the Plasma-Induced Magnetic Fields

Plasmas also induce magnetic fields due to the net movement of charges in their volume. In contrast to their electric counterparts, these magnetic fields readily penetrate biological materials at both low and high frequencies. This forms a motivation to consider their biological effects, in particular for tissue engineering and regenerative medicine. As supported by several experimental investigations, static magnetic fields have, for instance, effectively been used in bone unification, pain reduction and soft tissue edema treatment at surprisingly low levels [247,248,249,250]. Moreover, they can modify the microcirculation and possibly even the vascular tone in tissues [251,252,253], and additionally present an anti-tumor function [254]. Likewise, low frequency sinusoidal magnetic fields up to 300 Hz have been shown to enhance nerve and bone regeneration [255,256,257], to protect ischemic tissue [258,259], to reduce inflammation [260] and to exert anti-cancer effects [254]. Among the underlying cellular mechanisms, several empirical reports mention the enhanced proliferation and differentiation of non-malignant cells, the altered differentiation and inhibited proliferation of cancer cells, Ca^2+^ influx regulation, hormonal changes and the control of growth factor and ROS signaling [254,255,256,261]. Next to the therapeutic results, however, some adverse effects have also been reported [248,252,261,262].

Although these findings may serve as an inspiration, such static and low frequency fields are not expected under usual plasma application conditions, neither during nor after the treatment. More relevant to the present perspective article are the quickly fluctuating magnetic fields generated by the short-living micro-discharge filaments in DBD and the plasma bullets from atmospheric plasma jets. For example, the strongest magnetic field of a streamer propagating in ambient air is located precisely on its radius and at its head, with a magnetic field strength of about 1.7 × 10^−3^ T [263]. According to the measurements by Wu et al., the maximum magnetic field signal radiated by a helium plasma jet at standard conditions reached up to 0.055 dBm, agreeing with a current peak value of 75 mA carried by the plasma bullets and thus a transitional field strength of about 3 × 10^−5^ T at the plume edge [264]. This lies in the lower range of the orders of magnitude from 10^−6^ to 10^−1^ T usually applied for pulsed magnetic fields in experiments and clinical treatment [265,266]. Similar to the static and sinusoidal variants, pulsed magnetic fields are used for the treatment of musculoskeletal disorders, cerebral ischemic stroke, injured tendon, bone and nerve tissue, inflammation and cancer [254,256,259,260,265,266,267,268]. Treatment times can be as short as a few minutes [259], making it also comparable with plasma treatment in this regard [15]. The effects may remain subtle during brief plasma contact, considering that the clinical application times of pulsed magnetic fields typically span from tens of minutes to several hours, repeated over many days [259,265,266,268]. Nonetheless, a synergy can exist with the plasma-generated redox chemistry and electric field, so the magnetic field may still impact the biological response in a significant way. As should be kept in mind, pulsed electric fields and pulsed magnetic fields always have both an electric and a magnetic component, making a synergy between the two possible in all cases.

## 4. The Search for Synergies between Plasma and Nanomedicine

### 4.1. Enhancing the Strengths and Overcoming the Limitations

Synergies between plasma and nanomedicine are envisioned either by enhancing their strengths or by covering each other’s limitations. Such a proposal has already been put forward by Rasouli, Fallah and Bekeschus in their review on cancer treatment [41]. Yet, a comprehensive strategy is still lacking, in particular with regard to tissue regeneration and engineering. In this Section 4, we therefore lay out a strategy to further explore such synergies. The current Section 4.1 departs from some generic strengths and limitations of the technologies, in order to deduce a few synergies on a general level. As a second plan of action taken in Section 4.2, insights can be translated from the plasma medicine subdomain where the synergistic effect of NPs has already been actively investigated, i.e., in cancer research. Next, in Section 4.3, we briefly survey some lessons learned from the synergies of biotechnology in plasma medicine, and how they may apply to nanotechnology as well. Section 4.4 deals with the current knowledge on surface modification by plasmas, and its relevance to NPs in a liquid environment. Finally in Section 4.5 and Section 4.6, we build further on the knowledge and insights gathered in Section 2 and Section 3, to picture the redox- and field-related synergies, respectively.

In the context of cancer treatment, Rasouli, Fallah and Bekeschus have indicated the tumor morphology complexity and NP toxicity as the main challenges in nanomedicine [41]. For plasma technology, in contrast, they see the complicated plasma dose control and low RONS penetration depth into biological tissues as major obstacles. This line of thinking can be extended to regenerative medicine in a straightforward manner, because the morphology complexity, nanotoxicity, intricate plasma dose regulation and poor RONS penetration applies to the treatment of any type of injured tissue. However, the complexity issue can be understood as having both a negative and a positive side. On the one side, treatment optimization is obscured by the intricacy of various wound and tissue types, next to the cumbersome modulation of specific NP and plasma features. On the other side, the versatility of the nano- and plasma technologies permits a wide range of treatment conditions that can be tuned to each specific application. Combining them into a hybrid technology further amplifies this range of possibilities.

For a specific target, for instance, CAP is an ideal candidate for the combinatorial treatment with NPs, because the outcome of CAP sources can easily be tuned by changing its input parameters, such as voltage, electrode, feeding gas, type of power supply and the gap between plasma nozzle and target [18,99,108]. In our previous study, we illustrated this by means of the high amount of RNS generated with HNO_3_ vapor in comparison to water vapor with otherwise exactly the same device settings. Consequently, HNO_3_ vapor shows a higher antibacterial effect on *E. coli* bacteria than water vapor [18]. Similarly, water vapor as the feeding gas of the so-called COST plasma jet generates a high amount of •OH, significantly increasing the cell cytotoxicity of glioblastoma cells in the tumor microenvironment [269].

Nanotechnology, on the other hand, can be administered in the form of NPs and nanostructured scaffolds, available in different sizes, shapes, arrangements and materials [270]. As we know already from Section 2, the material has a crucial function in the biological response. In general, the physicochemical properties of NPs determine their tissue biodistribution, while their coating determines the receptor-mediated cellular uptake (Figure 8) [271]. When NP-bound ligands interact with a receptor on the cell membrane, the Gibbs free energy is locally decreased. This causes the membrane to wrap around the NP to form a closed-vesicle structure (see Figure 8b). Because the ligand density depends on the NP’s local curvature, the latter’s shape determines the internalization rate, as shown in Figure 8c. A high aspect ratio seems to cause a better biodistribution, e.g., more efficient tumor targeting and longer circulation, where the uptake rate depends on the angular orientation relative to the cell membrane [271,272]. Whereas rod-shaped NPs display a more rapid internalization than spherical ones for sizes above 100 nm, this order is reversed for sub-100-nm NPs [270]. NPs below a size of 12 nm were found to invade tumors more easily, likely because larger particles require more energy to be actively endocytosed [271]. Above 50 nm, a local depletion of ligands or receptors can occur, counteracting the receptor-mediated endocytosis [270]. Additionally, the NP composition strongly affects the endocytosis rate, e.g., with a 1000-fold difference observed between gold-NPs and single-walled carbon nanotubes, each 50 nm in diameter [270].

Despite these trends, a growing consensus has emerged among researchers that there is no one setting that fits all, for instance regarding plasmas or NPs to target cancer tissue [271]. The same conclusion should remain valid for regenerative medicine, considering the various types of tissues that can receive an injury and require healing. Looking at it from the positive side, both plasma and nanotechnology seem feasible to aid in the healing process of basically any tissue type, no matter whether it is epidermal [15,273,274,275], neural [46,141,276,277], osseous [45,276,278,279], etc. By combining both technologies, the treatment possibilities obviously expand tremendously. For example, as an interesting research question that has not yet received much attention, one can wonder how suitable plasmas and nanomaterials combined are for the treatment of specialized tissues, such as organs, which are hard to regenerate or engineer otherwise. This may hold a unique synergy on its own. However, this large degree of freedom prevents clinical adoption, unless standardized protocols are agreed upon. A customized protocol seems recommended for each application, i.e., for each tissue and injury type. Finding a viable balance between the various applications at the one hand and the number of standardized protocols on the other will be crucial to bring plasmas and nanomaterials into practice as a combined biomedical technology.

Next to this attractive parallel between plasma and nanotechnology with regard to their versatility, their complementary features deserve extra emphasis. As mentioned already in Section 2, plasma treatment provides an elegant solution to the unspecific NP toxicity, by permitting a lower dose of the latter while obtaining similar beneficial effects. Reversely, nanomaterials may facilitate the plasma dose control and aid the penetration and delivery of plasma-generated RONS, to overcome some of the main drawbacks of CAP in biomedicine. On its turn, CAP might allow regulating NP delivery in various ways, by means of the plasma-generated fields, pH change or a multitude of biomolecular pathways, as further discussed in Section 4.2.

In an experimental study, Zinovev et al. compared the efficiency of CAP and biopolymerous coatings in wound dressings for the treatment of third-degree skin burns [280]. According to their results, CAP treatment accelerates the healing process by 20% with a 52.5% in scar area reduction, but it is insufficient on its own. The use of wound dressings based on aliphatic copolyamide and chitosan nanofibers without CAP, in contrast, resulted in complete healing, accelerating the process by 42.8% with a scar area reduction of 65%. As such, they recommend combining both methods in future research. This reveals yet another important complementary aspect between nano- and plasma technology: nanomaterials can provide mechanical support to recovering tissue, while plasmas can supply additional redox- and field-related stimuli.

### 4.2. What Can We Learn from Cancer Research?

Nearly no research has been performed on the combination of plasma and nanotechnology for tissue regeneration specifically. However, the experimental data on the combination in the context of cancer research are growing quickly. Therefore, we hereby briefly map the state-of-the-art in this area and discuss which insights we can obtain from it in relation to regenerative medicine.

The intended combination of CAP and NPs leads to a synergic effect and can overcome each other’s limitations (Figure 9). For example, combining CAP with NPs reduces the required NPs concentration and modulates oxidative stress-mediated signaling, as mentioned in Section 4.1 and further discussed in Section 4.5. Such combinations have previously shown promising outcomes in drug transport activity, tissue regeneration, immunomodulation activity, and selective cancer treatment. Both technologies are directed to increase NPs uptake and RONS production regarding the mechanisms and potency. For example, in a study by Ouf, El-Adly, and Mohamed, silver-NPs as anti-dermatophytes were found to be the most promising agents against five dermatophytes in a combination with CAP. The optimal dose of silver-NPs is found to be much lower than that of anti-fungal drugs for skin disease [281].

The currently available studies on CAP conjugated with NP treatment in cancer research are listed in Table 2. Kim et al. reported that combining the CAP impact with antibody-conjugated gold-NPs modulates the cytotoxicity in melanoma cells by nearly five-fold [102]. Zhu et al. showed that CAP conjugate with drug-loaded core-shell gold-NPs substantially inhibited breast tumors [282]. Other studies have also suggested that iron oxide (Fe_2_ O_3_) has synergistic tumor reduction when conjugated with CAP in tumor treatment [283]. Further, the epidermal growth factor conjugated with gold-NPs shows significant DNA damage-mediated cell death after CAP exposure, which gives target-specific cell death in Epidermal growth factor (EGF) receptor-expressing cancer cells. However, the isolated treatment of epidermal growth factor conjugated with gold-NPs alone does not show a decrease in cell viability [284]. Therefore, for the combinatory treatment of plasma with NPs, more studies with the same treatment process and device are needed.

**Table 2 nanomaterials-12-03397-t002:** Combination of NPs and CAP treatment in different types of cancer.

Nanoparticles	Plasma Entity	Finding	Type of Cancer
Gold-NPs	Helium Plasma jet	Activation of intracellular ROS contents, cell cycle arrest, and intracellular anti-oxidant machinery leads to early apoptosis	myelomonocytic lymphoma [285]
	Plasma jet	Activation of intracellular ROS levels leads to cancer cell death	Glioblastoma [104]
		Cell death due to enhancement of intracellular RONS level and uptake of NPs	Brain tumor [102]
	Plasma jet	Cell death due to nuclear condensation and DNA fragmentation	Colorectal cancer [286]
	Gas Plasma	Decrease in cell proliferation and migration with induction of cancer cell death	Breast cancer [287]
		Production of intracellular RONS causes significant lipid peroxidation due to an increase in the uptake of gold-NPs through endocytosis	Brain tumor [104]
Anti-EGF receptor gold-NPs	Air plasma	Necrotic cell death	Skin and oral cancer [144]
EGF-conjugated gold-NPs		Selective apoptosis of cells having EGF receptor-mediated endocytosis	EGF receptor-expressing lung carcinoma cell [284]
EGF conjugated gold-NPs	DBD plasma	Increase the apoptotic response	Lung cancer [284]
Focal Adhesion Kinase antibody conjugated gold-NPs		Gap 1 (G_1_) phase cell cycle arrest leads to apoptosis	Skin cancer [109]
Gold-NPs		Stimulates clathrin-dependent endocytosis	Glioblastoma [144]
Polyethylene glycol (PEG) gold-NPs	Plasma jet	Singlet oxygen formation and formation of gold-PEG bond	Skin and oral cancer [288]
Fluorouracil (5-FU)loaded poly(lactic-co-glycolic acid) (PLGA) nanoparticles		induced down-regulation of metastasis-related gene expression	breast tumors [282]
Curcumin loaded on triphosphate chitosan NPs		Gap 2 phase/mitosis (G2/M) cell cycle arrest, upregulation of tumor suppressor markers	Breast cancer [289]
Iron-NPs		Tumor size reduction due to extensive necrosis	Lung Cancer [283]
PLGA-magnetic iron oxide		Triggering apoptotic process with DNA fragmentation	Lung cancer [112]
Platinum-NPs		Apoptosis due to cell cycle arrest, DNA fragmentation and augmentation of Ca^2+^ labels	Lymphoma [290]
Fluorouracil loaded PLGA NPs		Cytotoxic effects, metastatic gene expression reduction with the uptake of NPs	Breast cancer [282]

One of the most promising applications of these two technologies relates to drug transport. Plasma appears to increase the NP uptake across the lipid bilayer in the upper layers of skin via endocytosis [144] due to lipid peroxidation. He et al., showed that CAP stimulates the higher uptake of theranostic gold-NPs in glioblastoma cells, mainly via CAP-mediated lipid peroxidation at the cell membrane, which will be further discussed in Section 4.5 [144]. In our previous studies, we observed that lipid peroxidation increases the translocation of an antimicrobial peptide, Melittin (a major component of Bee venom), to a greater extent and cause severe cell death in skin and breast cancer [19]. Apart from that, CAP-generated oxidants can increase the uptake of insulin for cell proliferation [138]. This finding postulates that the plasma-generated physical and chemical entities are quite promising for the efficient transfer of NPs, as well as biomolecules such as DNA, proteins, and lipids [19,138]. For instance, localized temperature variation has also shown the delivery of NPs in preclinical studies [291]. Recently, Ankit et al. observed the on-demand delivery of silk-based carbon nanotubes for cancer therapy due to the triggering of chemo-drug release by a NIR laser [291].

Moreover, the synergistic effect of plasma technology with NPs is often observed by increasing ROS/RNS in the target tissue. For example, exogenous and endogenous ROS increased the penetration of lipophilic drug (curcumin) in plasma-treated murine and human skin [292]. In our previous work, we observed that the anti-cancer activity of plasma-treated liquid depends entirely on the plasma settings, especially on the plasma dose [108]. Recently, CAP and NPs have also attracted attention due to their intrinsic anticancer effects through activation of ROS production, resulting in DNA damage in the cancer cell. As should be noted, high intracellular ROS levels or oxidative stress can impair other intracellular components and cause apoptosis. Jawaid et al., showed that gold-NPs combined with He-CAP can significantly decrease cell viability by reducing intracellular antioxidant content such as glutathione. Further, failure of anti-oxidant machinery leads to the accumulation of intracellular ROS contents, which can cause total cell cycle arrest [285]. It is observed that the synergistic effect of NPs in combination with CAP also depends on the concentration of NPs [104], so thorough investigations are needed, not only for evaluating the plasma setup but also for their combined effects.

### 4.3. Exploring the Boundaries with Biotechnology

Nanotechnology in biomedicine can be distinguished from biotechnology as a whole, in the sense that the former relies on artificially synthesized materials, whereas the latter usually relies on biologically derived materials. Yet, this distinction may be seen as somewhat arbitrary, because it both concerns nanostructured materials, which mainly differ from each other in their origin, rather than their envisioned function. Indeed, biological macromolecules act as nanomaterials in nature, with properties closely related to their small size. For this reason, several similarities are expected in their behavior in a biological environment, such as an injured tissue.

Moreover, after billions of years of biological evolution, it is now firmly believed that organisms have developed themselves into a considerably optimized state. This natural advancement serves as inspiration for scientists and engineers who aim to improve certain inventions with analogous purposes or functions. This foundational idea of biomimicry also applies to the building blocks of such inventions at the one hand and of living organisms at the other, especially at the atomic scale. This gives a second reason to study the behaviour of biological nanomaterials, in order to gain insight into their artificial variants.

Applying this line of thinking to an envisioned synergy between CAP and NPs in regenerative medicine, one can find inspiration in the available knowledge on the combination of CAP and biotechnology. Although we expect this strategy to have a very wide applicability, it is not our ambition to be fully comprehensive on this topic. Instead, we will provide here three examples.

As the first one, we refer to the high potential of stem and progenitor cells in plasma medicine, with tissue engineering and regeneration in particular. These refer to a specific group of cells characterized by their abilities of self-renewal, multipotent differentiation, and repair after organ injury [293]. As an important difference, progenitor cells possess a limited capability of differentiation in comparison to stem cells. Apart from the promising combination of stem and progenitor cells with nanomaterials, synergies have also been observed with plasma treatment. CAP can regulate stem cell fate both directly and indirectly, by stimulating the cells in close contact and by modifying the cell-resident niche, respectively [294]. More specifically, CAP promotes stem cell proliferation by means of multilevel events, such as a faster cell cycle at the cellular level, activation of stem cell-specific markers at the protein level, and a beneficial deregulation of genes at the transcriptomic level. It also triggers or promotes stem cell differentiation into various tissues such as nerves, bones, and cartilage, likely through more complex mechanisms. For an elaborate discussion on the underlying biochemical mechanisms, we refer to the review by Tan et al. [294]. Next to the biochemistry, we also know from Section 3.3 that stem and progenitor cells are prone to electric field-regulated biophysics. The plasma field penetrating into the tissue, as well as the charge deposited on its surface, enable several biological effects. Many of these effects are related to a modification of the microstructure, including cell alignment and directional transport. Especially these transport mechanisms may form a parallel between stem and progenitor cells at the one hand and NPs at the other. Therefore, finding these parallels can strongly enhance our understanding on how to control bio- and nanomaterial transport in vivo by means of common guidelines.

Besides stem and progenitor cells, various types of growth factors play a crucial role in regenerative medicine. These are naturally occurring substances, usually secreted proteins or steroid hormones, able to stimulate cell proliferation, wound healing, and occasionally cellular differentiation [295,296]. An interesting experimental study in this regard has been performed by Tan et al. on wound healing, angiogenesis, neurogenesis, and osteogenesis [297]. They used a multimodal treatment method with CAP and acidic fibroblast growth factors, finding a synergistic enhancement for the wound healing and angiogenesis. The former was demonstrated by increased murine fibroblast proliferation and reduced cutaneous tissue inflammation, whereas the latter by upregulated proangiogenic markers in vivo and downregulated antiangiogenic proteins in vitro.

As a third example, Przekora et al. [298] used a polysaccharide matrix of a bone scaffold made of a hybrid chitosan/curdlan matrix and hydroxyapatite granules to immobilize NPs and to support human adipose tissue-derived mesenchymal stem cells under plasma treatment. The plasma-generated ROS had a positive impact on stem cell proliferation, while not interfering negatively with their osteogenic differentiation. Although the study focused on the effect of the immobilized NPs, we want to draw attention to the modified biomaterial that functioned as the scaffold. In its own right, it may be regarded as a nanomaterial. Its behavior under plasma treatment is therefore relevant to nanomaterials considered as alternative scaffolds, or it can inspire the design thereof. As such, we want to motivate the plasma medicine community to search for synergies between CAP and biological or bio-inspired NPs or semisynthetic materials, such as chitosan, curdlan and hydroxyapatite, next to the usually applied nanomaterials of non-biological origin. That is, the use of such bio-inspired substances seems an intuitively appealing strategy to overcome the unspecific toxicity risk of purely synthetic nanomaterials.

### 4.4. In Situ Plasma Modification of Nanomaterials

Given the biological complexity of tissue, some NPs are not biocompatible, which is the biggest challenge of nanomedicine [299,300]. Hence, CAP might be an option to reduce the problem of NP biocompatibility. For example, CAP is widely used to improve the biocompatibility and bio-functionalization of synthetic biomaterials and to produce several exclusive properties compared to other chemical and physical methods. Most commonly, this is accomplished by treating the material surface under direct plasma contact. Plasmas can namely modify surfaces in the gas phase through etching, deposition, nanostructuring and functionalization. In this way, plasma treatment allows to modify the surface roughness and to increase the hydrophilicity of biodegradable polymers, thus having great potential for applications in dentistry, tissue engineering, and improving cell affinity and adhesion [301,302]. More specifically, plasma processing has often been shown to promote the adhesion of stem/progenitor cells to foreign materials, such as NPs and nanostructured scaffolds, useful for regenerative medicine. Similarly, the attachment, loading, and release of drug molecules in porous biomaterials can be improved with plasma surface modification [303], which is due to the resulting structure as well as chemical modification. Ueda et al., for instance, observed the attachment of stem cells after modifying the polystyrene surface by plasma treatment [304]. Further, alterations of surface chemistry of polydimethylsiloxane by plasma-generated reactive species favor the adhesion of human mesenchymal stem cells in bone tissue regenerations [305].

Additionally, plasma technology has been widely used in the fabrication and synthesis of nanomaterials. Plasma-generated reactive species can, for example, directly reduce metal ions in the liquid phase to form metal NPs, without the presence of any additional reducing agents [6]. This suggests several synergies between plasma medicine and nanomedicine. Plasma treatment should therefore, in theory, enable the activation, modification, and even decomposition of nanomaterials present in biological suspensions or tissues. Scientists and engineers working in plasma and nanomedicine will therefore benefit from a more profound insight into the underlying surface modification mechanisms. However, most of the research on plasma surface modification deals with direct plasma–surface interaction in a gaseous environment. For nanomaterials in plasma medicine, the mechanisms in the liquid phase are more representative for the in situ processes at their surface under plasma treatment. For this purpose, it will be useful to gain more knowledge on, and insight into, the redox chemistry at surfaces and NPs in biological solutions under plasma treatment, a topic that is yet relatively unexplored. Understanding this chemistry may lead to the discovery of new synergies between plasma and nanomedicine, enabling further optimization.

### 4.5. Redox-Related Synergies

As demonstrated with the insights from Section 2 and Section 3, plasma and nanotechnology each provide unique building blocks and administration methods for tissue regeneration and engineering. Despite their obvious differences, a few parallels are readily identified. For instance, several types of NPs are able to produce a redox chemistry in a biological environment, leading to a broad spectrum of oxidation-mediated biological pathways. The same pathways are under consideration in plasma medicine, where research is heavily focused on redox-related effects. As the main difference, most NPs can produce only endogenous ROS, whereas plasmas provide reactive species exogenously, including RNS, which can then trigger cell signaling pathways that create endogeneous ROS. This implies a first possible redox-related synergy between the two technologies.

Studies on nanotechnology in biomedicine that involve RNS are rather scarce. As an exception, nitric oxide (NO)-releasing NPs or nanomaterial platforms have received considerable attention in regenerative medicine, due to the powerful beneficial biological function of NO in wound healing and infection treatment [306,307]. During the past four decades, NO has been recognized as one of the most versatile participants in the immune system. It plays an important role in the pathogenesis and control of tumors, infectious or chronic degenerative diseases and autoimmune processes [308]. As a short-lived gas, its application has, however, been limited by various practical shortcomings, related to the short storage and release times, in addition to the poor controllability of the temporal and spatial delivery [309]. NO-releasing nanotechnology has therefore been investigated as a versatile alternative. Just like all other nanotechnologies, it has its own limitations. CAP provides a solution to the storage and release issues of the external gas application, while being complementary to the nanomaterials in various ways. It allows, for example, on site production of NO, which can subsequently be distributed by means of nanovehicles. Additionally, it can supply other types of RNS, which can be transformed into NO through nanocatalysis. In this way, NPs no longer need to be synthesized with NO sources to take part in the NO biochemistry, which permits a more versatile and safer NO-based treatment.

This reasoning can essentially be extended to any type of RONS. In short, we propose to generally distinguish the following redox-related aspects on which synergies can be based:In line with Section 4.4, plasma-generated RONS can modify and consume the nanomaterial surface at a controllable rate. This provides a means to vary the release rate of NP matter, to regulate the location where such release occurs and/or to completely degrade a nanomaterial in situ, all very useful abilities to restrict short- and long-term toxicities.Reversely, CAP is, in theory, able to supply or resupply nanomaterials with RONS, for instance by an adhesion or deposition process at their surface. In other words, nanomaterials can function as RONS batteries that can be recharged by means of plasma treatment. This enables an interesting flexibility, since such nanomaterials may be neutral to the redox chemistry in their completely discharged state, and charge selectively or unselectively with regard to different RONS. Accordingly, the CAP parameters can be used to vary the RONS to be released in a reversible manner, without the need to remove or replace the nanomaterial.Analogously, nanomaterials may serve as RONS regulators or catalysts without being consumed, loaded or having their surface conditions altered. Inspiration on such mechanisms can be taken from the large body of knowledge on plasma catalysis [310,311,312]. However, similar to our remark on plasma surface modification in Section 4.4, plasma catalysis is mainly considered in the gas phase. Therefore, the available knowledge on heterogeneous (nano)catalysis in the liquid phase may be taken instead from the literature on advanced oxidation processes used in water treatment, in particular the plasma technology used for this purpose (see e.g., [313,314,315]). Another source of inspiration can be found in the available knowledge on the antioxidant and ROS-generating mechanisms for various NPs (Figure 10 and Figure 11).

The plasma-induced redox chemistry can be used to modify the nanomaterial’s liquid environment, for instance in terms of pH and conductivity. In this way, the solubility, chemical activity or electrical behavior of the material becomes tunable in situ.Further, the combination of plasma and nanotechnology allows a dual enhancement in delivery mechanisms. On the one hand, NPs may serve as nanocarriers to transport plasma-generated RONS. On the other hand, CAP may modify the liquid medium to facilitate NP transport. A particular example is given by He et al., who found CAP to stimulate clathrin-dependent endocytosis, due to a repairment mechanism of oxidised cell membrane [144]. This enhanced the uptake of nanomaterial in glioblastoma multiforme cells.As mentioned on several occasions above, combining plasma and nanomedicine permits lowering the applied dose, while enhancing the treatment selectivity. With relation to RONS, each of the aforementioned points can be employed for this purpose.

However, these envisioned synergies should still be considered speculative, as experimental evidence for them is largely lacking for now. We therefore hope to stimulate more fundamental empirical research, aimed toward validating these ideas.

### 4.6. Field-Related Synergies

In analogy with the redox-related synergies, a parallel can be identified between plasma and nanotechnology with regard to the role of electric and magnetic fields. As discussed in Section 3.3 and Section 3.4, the plasma-induced fields are preferably distinguished into three groups: (i) short-lived electric fields that penetrate into the tissue due to their rapid fluctuation, (ii) long-lived DC electric fields induced by the deposited charge, and (iii) the magnetic field components. As a first possible synergy, nanomaterials with electric or magnetic properties can aid in the delivery of these fields deep into the tissue under treatment, in order to invoke one of the direct field effects discussed in Section 3.3 and Section 3.4. Accordingly, also the selectivity and dosing of such treatment may be improved in this way. This is especially relevant to the short-lived electric field fluctuations with a small penetration depth under normal conditions without nanomaterials. Plasmonics and electric field enhancement are two ways to accomplish this. The use of electroactive nanomaterials for the enhanced delivery of electric fields has been discussed in several reviews [317,318,319].

As a second possible synergy, the plasma-induced fields may enhance the delivery of NPs, and correspondingly increase their selectivity and lower their required dose. This is expected in particular for the long-lived electric field from the deposited charge, because it retains a specific orientation and therefore allows bidirectional transport of charged species. Indeed, analogous to the bidirectional transport of cells, as discussed in Section 3.3, charged, polarized or even neutral nanoparticles can be moved through a liquid medium in a steady electric field through electro-osmosis, electrophoresis, dielectrophoresis or electromigration [320,321]. The same is true for fluctuating electric or magnetic fields, especially if they possess a non-zero DC component [322,323].

As a third possible synergy, the plasma-induced fields are able to activate nanomaterials in various ways, depending on the type of nanomaterial. For instance, piezoelectric NPs and nanostructured scaffolds deform under an electric field, which results in mechanical stress that on its turn can induce oxidative stress. As should be noted, however, these materials are often applied in regenerative medicine for the opposite purpose, i.e., to generate a local electric field under mechanical stimulation [32,319,324,325]. It is also worth noting that piezoelectricity plays a natural role in the growth, structural remodeling, and fracture healing of living bone tissue [326]. Other collagenous tissues such as tendon, cartilage, cementum and dentin have also been observed to display piezoelectricity [324,327].

Next to that, magnetoelectric materials generate a local electric field under influence of a magnetic field. Because of this ability, magnetoelectric nanocomposites are considered as alternatives to their piezoelectric counterparts in regenerative scaffolds [32]. Interestingly, they also exhibit a magnetomechanical response and can become magnetized through electrical stimuli. Under plasma treatment, they can therefore react in various ways useful to tissue regeneration.

As mentioned already in Section 2, magnetic NPs are also used in regenerative medicine and tissue engineering. The transparency of biological tissue to magnetic fields enables the straightforward control over these NPs, which makes them especially attractive for drug delivery and selective treatments [328,329]. Moreover, magnetic composite scaffolds form a means to provide local mechanical stimuli to cells under an external magnetic field. Correspondingly, magnetic nanomaterials have been shown to stimulate cell adhesion, proliferation, and even differentiation, which are essential to tissue regeneration and engineering [328,329,330]. A synergy may therefore also be possible with the magnetic field component delivered by CAP treatment, provided that it is intense enough. However, several concerns have been reported on the toxicity of these materials, with conflicting results between in vitro and in vivo studies [329,330].

In summary, plasma and nanomaterials may display synergies through distinct effects, such as mechanical responses, field enhancement, chemical effects and mass transport. This provides an appealing multidimensional playground for researchers working in plasma and nanomedicine, for instance to obtain an envisioned effect in tissue engineering or regenerative medicine. Nonetheless, as on any playground, safety rules are required. This will be discussed in Section 5.

## 5. Safety Concerns

Biomedical applications of NPs have a long and rich history, but despite their rapid advancement, a prolonged exposure at specific concentration levels causes adverse health effects in organisms and humans [299,300]. Even on the cellular level, the toxicity of NPs also induces a high expression of inflammatory cytokines and chemokines, which further hamper the normal cellular functions and physiopathologic outcomes, including genotoxicity, apoptosis, necrosis, fibrosis, metaplasia, hypertrophy, and carcinogenesis [331,332,333]. The toxic potential of NPs depends on their size, shape, charge, functionalized groups, and free energy, which determine their tendency to cross the cellular membrane [333]. Most NPs have a high affinity to bind with biomolecules. This leads to adverse biological outcomes through protein unfolding, loss of enzymatic activity, DNA adduct formation, fibrillation, and thiol crosslinking [334,335]. Another issue is the release of toxic ions by NPs when thermally conductive materials conjugated with them favor dissolution in a biological microenvironment.

Additionally, the selective NP delivery to targeted tissues is one of the prime concerns. Since NPs can circulate in the blood for a prolonged period of time, they may be distributed non-selectively in neighboring tissues, rather than accumulate in the region of interest [335]. Thus, NPs in current biomedical applications are often used in an adapted combination therapy with controlled drug delivery, leading to less toxicity. To this end, plasma technology in combination with NPs is envisioned as a mainstay strategy to overcome the limitations, as shown in various published articles. Based on these studies, the synergy between these two technologies will improve the NP delivery with optimum concentration. Importantly, CAP can be used to create a controlled biological environment adapted to the multifunctional and multimodal nature of the NPs in relation to their surroundings, in order to make them bind with the targeted tissue only.

A similar remark applies to plasma treatment on its own. Although plasma medicine has brought forward several remarkable achievements, such as cancer therapy, chronic wound therapy, and antimicrobial effects, the selective treatment of targeted tissue, such as a tumor or an injury, remains challenging. It is observed that the plasma-generated reactive species are consumed significantly by the actual biological tissue. A total of 5% of the ROS and around 80% of RNS can penetrate across only 500 microns of tissue [336]. Therefore, plasma needs to be combined for the treatment of deep-tissue chronic wounds. For this purpose, the combination of plasma-generated ROS/RNS and electric fields with NPs could be a precise and selective therapy, to obtain a desired dose distribution.

As another challenge in plasma medicine, various plasma sources have been developed by numerous research groups. This has led to the absence of a detailed framework for device characterization, as one of the most critical issues in the field. Moreover, a multitude of operating conditions is involved during plasma processing, making it hard to specify the optimal plasma exposure time and energy delivery during exposure. Some groups have shown a decreased cell migration rate at low plasma exposure [337,338,339], while others reported enhanced cell migration at a similar treatment dose but with a different plasma reactor design [337,338,339]. This lack of comparability between different devices and their biological effects remains a major problem to successfully transfer plasma technology into clinical practice.

In addition, long-term clinical effects of CAP are still largely unexplored. In one study, it has been shown that continuous and long-term plasma exposure can induce mutagenicity or genotoxicity [340], which would limit the implementation of plasma technology in the clinic. Hence, the correct dosage and exposure time also need to be considered in the treatment design. To overcome these discrepancies in plasma technology and promote successful outcomes and applications, a strict evaluation, along with an open discussion on its potential risks and restrictions, is required. This is not only related to the safety concerns, but also to assist the adoption of this novel technology by clinicians, healthcare providers and patients, as well as to promote overall public acceptance. This once more emphasizes the need for standardized protocols.

Whereas combining plasma and nanotechnology may partly resolve the above toxicity issues, any undesired or hidden synergies can also amplify the risks. NPs could, for instance, display an unintendedly high hazardous catalytic activity in the plasma-generated redox chemistry of its direct surroundings, while regulating it to a safe level elsewhere, or vice versa. Similarly, the plasma-induced electric or magnetic field might interact with the NPs in a different way locally and non-locally, for instance due to local field enhancement or inadvertent resonances. Additionally, CAP may amplify the toxicity of certain NPs in the short or long term, by enhancing the release of toxic ions or other constituents from the NP, or by decomposing them into dangerous by-products with possibly an elevated cross-toxicity. Synergies between plasma and nanotechnology therefore form a double-edged sword. As such, investigating them is not only prudent for a further optimization of the envisioned biological effects, such as wound healing or tissue engineering, but also crucial to the safe advancement of the combined technology.

## 6. Conclusions

Nano- and plasma medicine have emerged independently from one another during their initial stages, in different time frames. Nanomaterials in biomedical applications can for a large part already be considered a mature technology, whereas the use of cold plasmas still mostly resides in its preclinical phase. Only recently, the first experimental trials on their combination have appeared, mainly in the context of cancer research. Anticipating on the same trend in regenerative medicine, this perspective article explores the possible synergies between the two technologies, with particular attention to wound treatment. For this purpose, we first focused on the known underlying mechanisms at work in each technology individually, to envision subsequently how they can interfere with each other in a combined setting. We note that, on the one hand, many types of nanomaterials can activate cells through oxidative stress-mediated pathways. In particular, nanoparticles or nanostructured scaffolds with sophisticated electrical and mechanical features allow regulating the endogenous redox and biochemistry by means of electrical and mechanical stress. On the other hand, the plasma-induced redox chemistry, electric and magnetic fields each act as distinct stress factors with similar biological effects. These stimuli display different thresholds above which these mechanisms are activated, characteristic to each class of stimuli, each mechanism, each cell type and its environment. As a convenient coincidence, they generally allow to deactivate pathogens or undesired cell lines such as cancer cells in a certain intensity range, while leaving healthy tissue intact. It is therefore important to quantify the oxidant and electromagnetic field concentration in cells, as well as to keep the induced perturbation within the defense network regime of the untargeted normal cells.

The analogous importance of redox- and field-related stressors for nanomaterials and cold atmospheric plasmas implies several possible synergies between the two technologies in biomedical applications, especially with regard to regenerative medicine and tissue engineering. Accordingly, we propose several plans of action to further explore these synergies. First and foremost, synergies can be found either by enhancing the strengths or by overcoming the limitations of the two technologies with one another. In particular, the combination permits a lower dose of each stressor, which significantly reduces the related toxicity risks. Besides that, nanomaterials can be used as mechanical support for injured tissue, a complementary function that cold plasmas lack. Secondly, current insights into the synergistic processes at work in the context of cancer treatment seem to a certain extent translatable to regenerative medicine. This seems especially useful in order to find parallels in transport mechanisms, for instance in extracellular drug delivery or transmembrane transmission. As a third strategy, the synergistic combination of cold plasma treatment with biotechnology may also serve as inspiration for new developments. This includes research on stem and progenitor cells, growth factors and other biological, bio-inspired or semi-synthetic nanomaterials into the picture.

Fourthly, the plasma-induced surface chemistry of nanoparticles and nanostructured surfaces in the liquid phase is expected to hold several synergies, for which a fundamental understanding may be based on plasma surface modification research performed for other applications. However, most of the existing research on plasma surface treatment focuses on the processes in a gaseous environment. More investigations on RONS-mediated surface modification in a liquid is thus desirable. Fifthly and sixthly, a profound insight into both the redox- and the field-related fundamental processes also opens the door to the optimization of known synergies or the discovery of new ones. Research in plasma medicine is currently heavily focused on the redox biochemistry, whereas the field-related biophysics has received considerably less attention up to now. We therefore want to motivate the plasma and nanomedicine communities to explore the latter as well. This approach namely seems to hold a very high potential, considering the proven function of naturally occurring endogenous electric fields in biological processes on the molecular, cellular and tissue level, especially in wound healing.

Finally, we emphasized the toxicity concerns in plasma and nanomedicine, which may be partly remediated by their combination, but also partly amplified. Unintended and hidden synergies therefore need to be identified and profoundly understood, not only for a rapid, but also for a safe advancement of the combined technology. Using standardized protocols and materials is therefore strongly recommended. It will deepen our insight into the induced biological mechanisms for well-characterized device settings and experimental conditions, while simultaneously ensuring a faster clinical adoption of this very promising hybrid biotechnology.

## Figures and Tables

**Figure 1 nanomaterials-12-03397-f001:**
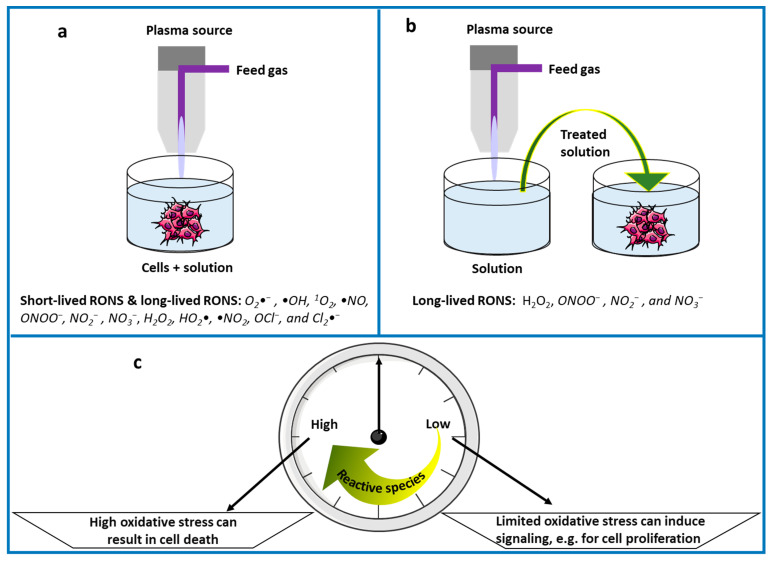
Schematic illustration for the use of CAP in (**a**) direct treatment and (**b**) indirect treatment, indicating also the typical “products” of both strategies, i.e., both short-lived and long-lived RONS in (**a**) and only long-lived RONS in (**b**). (**c**) shows that low levels of reactive species play an important role in cell signaling for various biological processes, such as cell proliferation, whereas high levels can induce cell death. Note that (**a**,**b**) can both give rise to high and low RONS levels, and thus to cell death and cell proliferation.

**Figure 2 nanomaterials-12-03397-f002:**
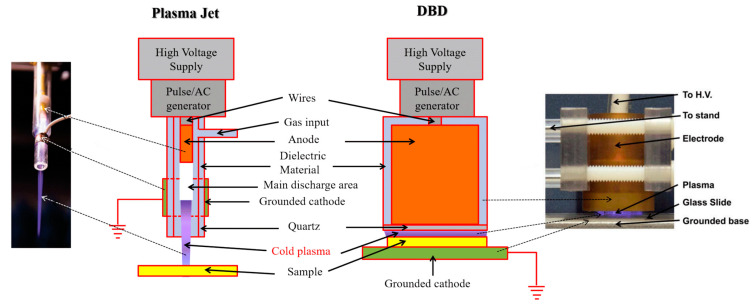
Schematic illustration of a plasma jet and a DBD. Reprinted with permission from Ref. [37]. © 2017 Yan et al.

**Figure 3 nanomaterials-12-03397-f003:**
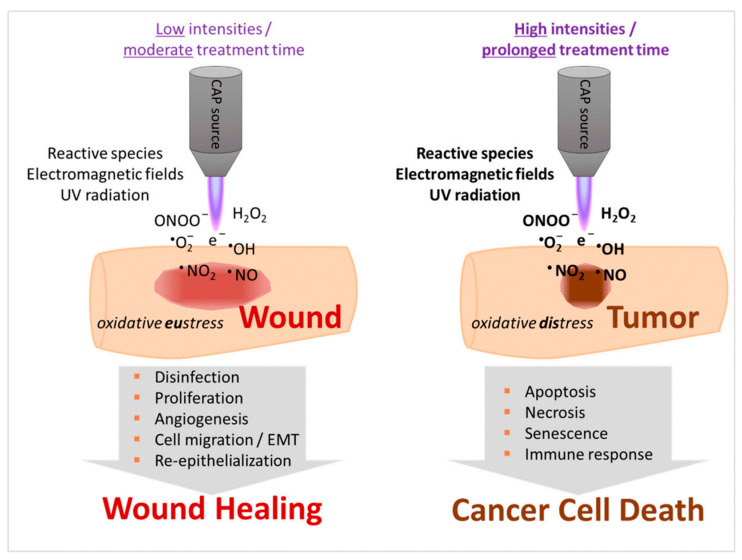
Comparison between CAP for wound and cancer treatment (**left**). Short or moderate treatment with a low intensity triggers biological processes essential to tissue regeneration, such as disinfection, proliferation, angiogenesis and epithelial-mesenchymal transition (EMT) (**right**). Prolonged treatment with a high intensity, in contrast, shifts the oxidative stress from eustress to distress, which is more suitable for cancer therapy. Reprinted from [15] under the terms of the Creative Commons Attribution License.

**Figure 4 nanomaterials-12-03397-f004:**
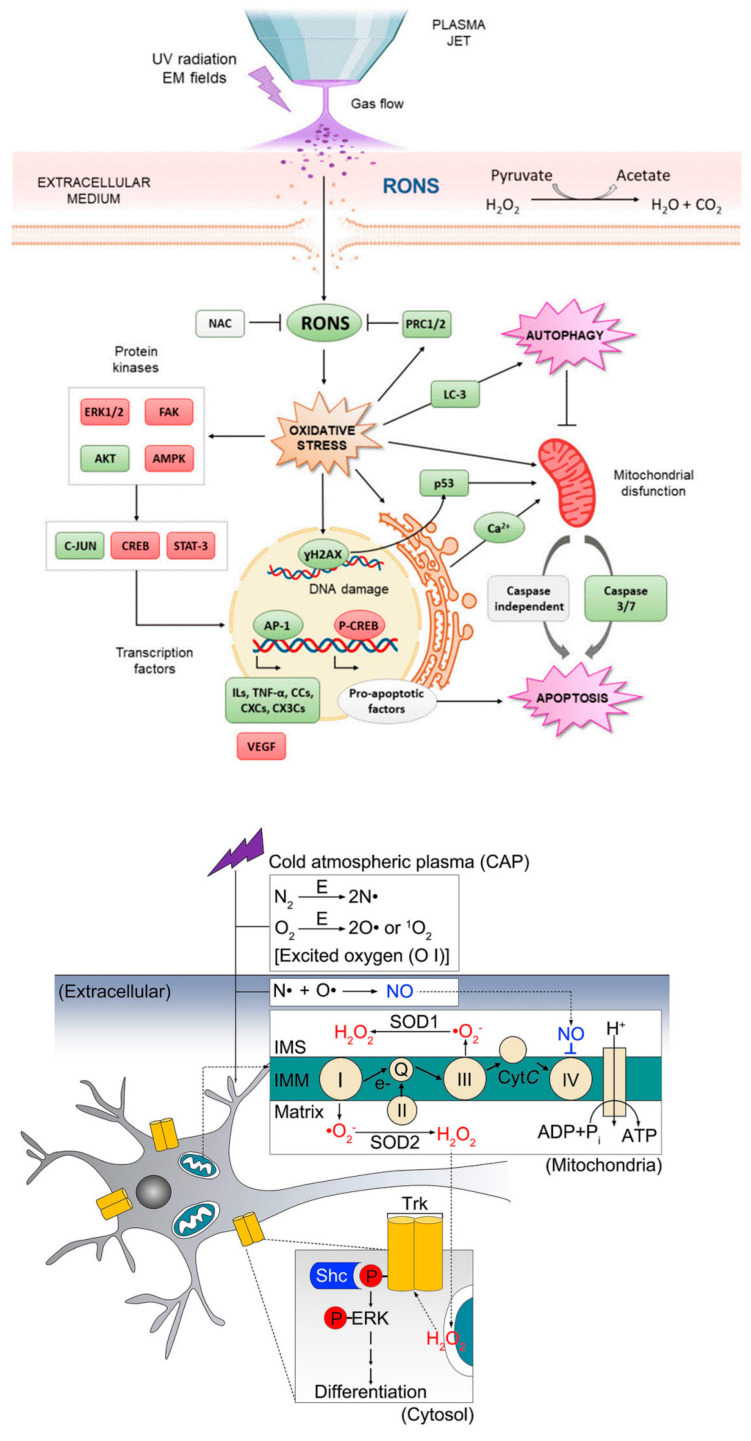
Plasma-induced oxidative stress-mediated signaling pathways proposed in the literature (**top**). In osteosarcoma therapy, according to Mateu-Sanz et al., pyruvate scavenges hydrogen peroxide in the extracellular medium. CAP-induced cell membrane disruption, however, facilitates internalization of RONS into the cell. The elevated oxidative stress increases peroxiredoxin expression and modulates protein phosphorylation and transcription factor activation, which dictates cell death or survival. Next to that, the plasma-generated RONS can change the expression pattern of cytokines, chemokines and growth factors responsible for osteosarcoma progression. CAP also causes DNA damage related to p53 activation and caspase-dependent apoptosis. Alternatively, mitochondrial injury may be inflicted directly by the RONS or indirectly by an increased mitochondrial calcium influx. Green boxes indicate increase or upregulation, whereas red boxes designate decrease or down-expression (**bottom**). According to Yan et al. following Jang et al., neural differentiation is initiated by the competition of NO with O_2_ for binding to the active site of the cytochrome c (CytC) oxidase, mitochondrial electron transport chain complex IV, and reversibly impeding its activity. Mitochondrial •O_2_^−^ is formed via electron reduction of O_2_ by obstructing electron transfer in the electron transport chain. Next, the mitochondrial •O_2_^−^ results in intracellular H_2_O_2_ generation, able to activate the Trk/Ras/ERK signaling pathway by acting as an intracellular messenger, ultimately leading to neural differentiation. (**top**) Reprinted with permission from Ref. [140]. © 2021 Mateu-Sanz et al. (**bottom**) Reprinted with permission from Ref. [141]. © 2019 Yan et al.

**Figure 5 nanomaterials-12-03397-f005:**
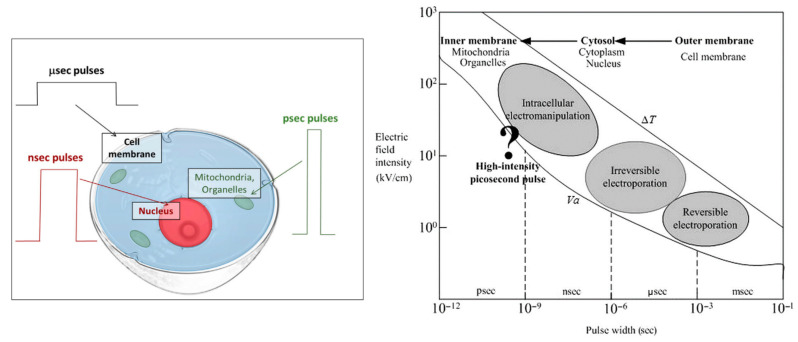
The biological effects of electric field pulses as a function of the pulse length and magnitude, with (**left**) the cellular structures preferentially affected by each type of pulse and (**right**) the resulting biological mechanisms. (**left**) Reprinted from [162] with permission from Elsevier, based on [165]. (**right**) Reprinted from [166] with permission granted from the editor.

**Figure 6 nanomaterials-12-03397-f006:**
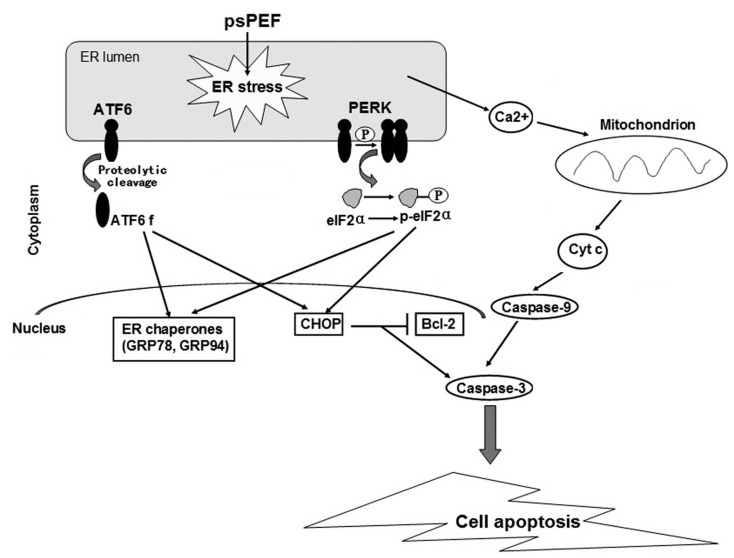
Scheme of the signaling network through which a picosecond pulsed electric field (psPEF) induces apoptosis in HeLa cells, as proposed by Chen et al. The pulses disturb the endoplasmic reticulum (ER) function, resulting in the release of Ca^2+^ and the depolarization of mitochondria, which in turn release cytochrome C (cyt c) and activate the caspase family. The ER dysfunction also causes two of its transmembrane proteins, activating transcription factor 6 (ATF6) and protein kinase-like ER kinase (PERK), to activate the unfolded protein response. The ATF6 pathway promotes the expression of ER chaperones and the pro-apoptotic transcription factor CCAAT enhancer-binding protein (C/EBP) homologous protein (CHOP). Phospho-PERK instead phosphorylates eukaryotic initiation factor-2α (eIF2α), which decreases gene expression. Reprinted from [170] with permission granted from the editor.

**Figure 7 nanomaterials-12-03397-f007:**
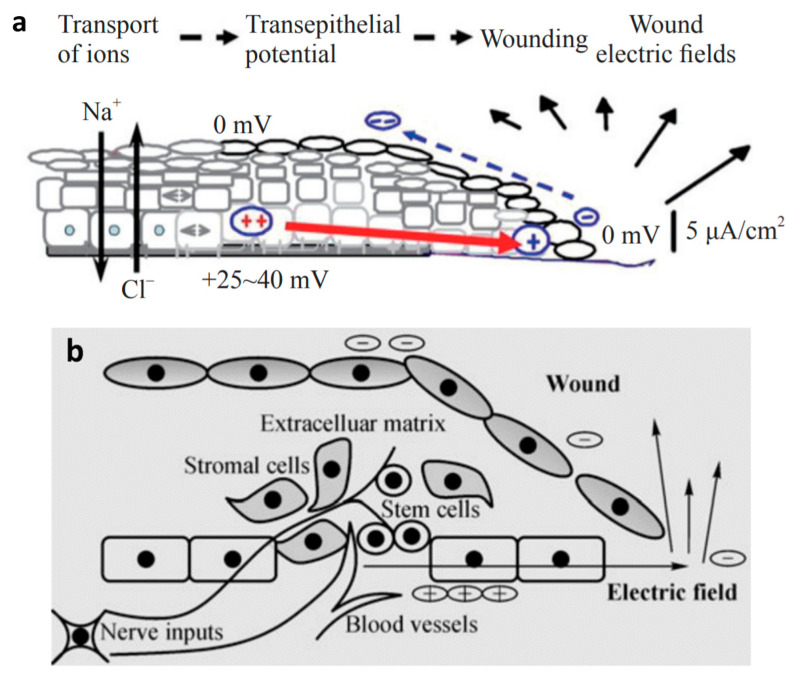
Schematic diagram of the electric field at a wound. (**a**) Na^+^/K^+^ ATPase pumps transport ions through the epidermis of intact tissue, resulting in the skin battery. At a wound, the transepithelial potential is short-circuited, inducing a lateral electric field pointing toward the center of the wound. Adapted from [236], with permission from Elsevier. (**b**) The model proposed by Li and Jiang for a stem cell niche in a skin wound, composed of stem cells, stromal support cells, extracellular matrix proteins, blood vessels and neural inputs. According to the model, endogenous wound electric fields exert effects on the local stem cells for days until the wound heals. Reprinted by permission from Springer Nature: [237].

**Figure 8 nanomaterials-12-03397-f008:**
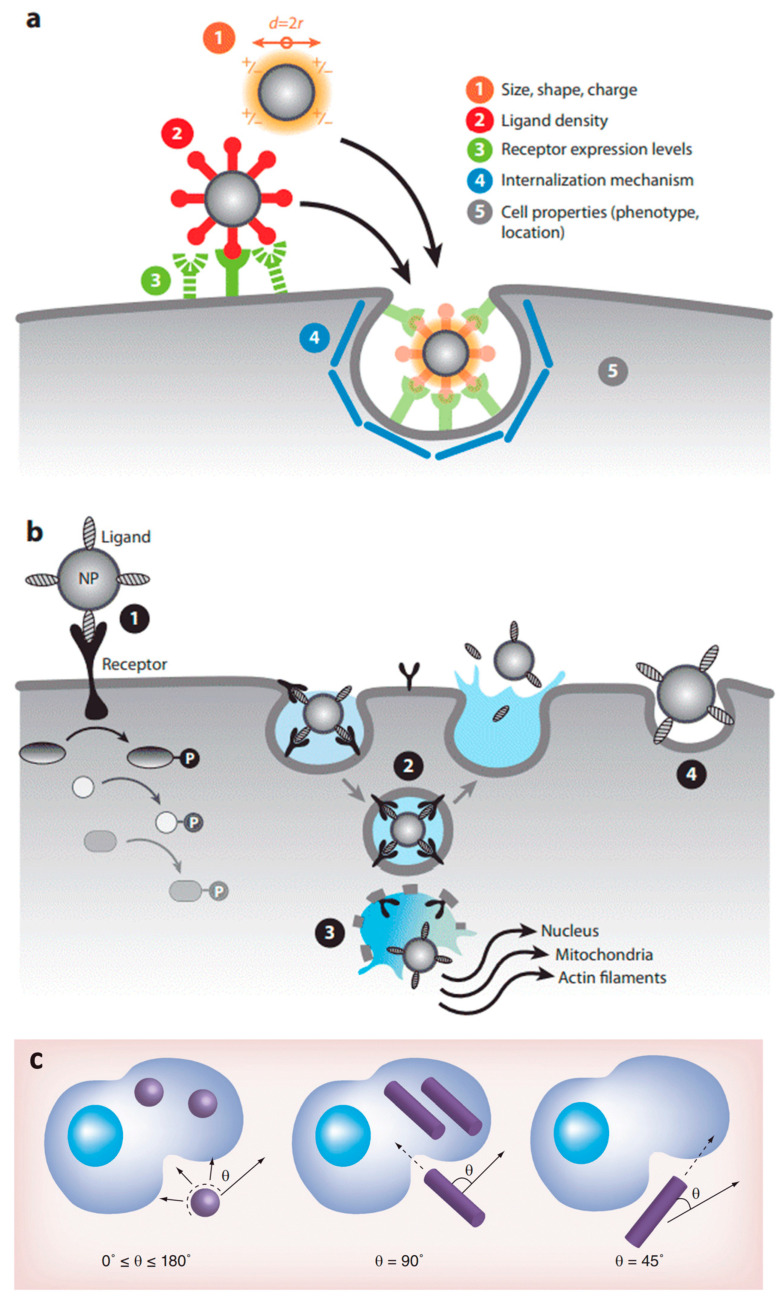
Internalization mechanisms for NPs into cells. (**a**) Factors affecting the internalization, related to the NP and cell properties. (**b**) Uptake mechanism of ligand-coated NPs, where (1) the NPs first bind to receptors on the membrane, inducing an intracellular signaling cascade while staying outside the cell. (2) The NPs may subsequently enter and exit the cell without ever leaving the vesicle; or (3) they can escape the vesicle and interact with different organelles. (4) They may also enter the cell after nonspecifically interacting with its membrane, i.e., not mediated by receptors. (**c**) Influence of the NP shape and contact angle θ on the internalization rate. The latter is independent of θ for spherical NPs due to their symmetry, whereas rod-shaped NPs internalize most rapidly with their major axis perpendicular to the cell membrane. (**a**,**b**) Used with permission of Annual Reviews, Inc., from [270]; permission conveyed through Copyright Clearance Center, Inc. (**c**) Used with permission of Future Medicine Ltd., from Reprinted from [272]; permission conveyed through Copyright Clearance Center, Inc.

**Figure 9 nanomaterials-12-03397-f009:**
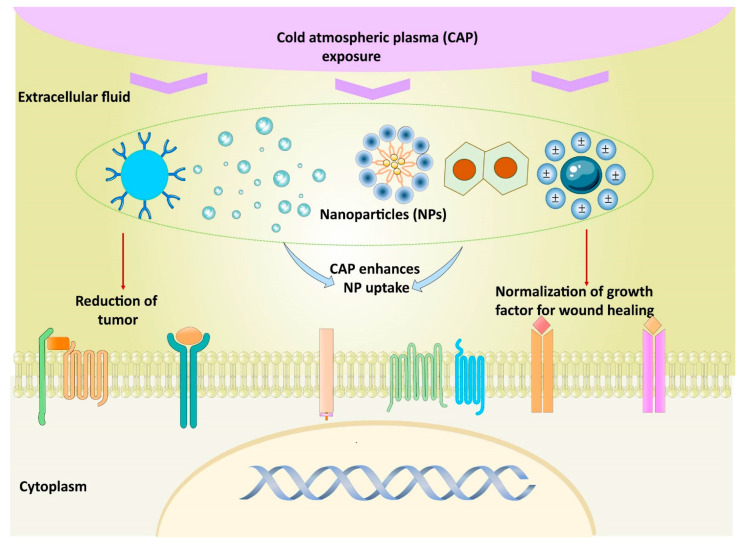
CAP provides a promising tool for nanomaterial-assisted tissue regeneration in plasma medicine. The symbol ± denotes the either positive or negative ions surrounding a NP with opposite charge.

**Figure 10 nanomaterials-12-03397-f010:**
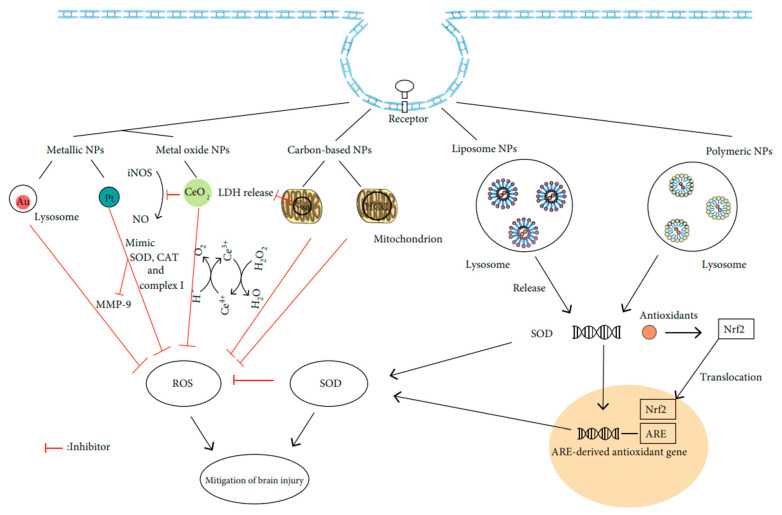
The antioxidant mechanisms of various NP types, after entering a cell by receptor-mediated endocytosis. Metallic, metal oxide and carbon-based NPs exert free radical scavenging properties, for instance by mimicing the activity of superoxide dismutase (SOD), catalase (CAT), and mitochondrial complex I, to decrease the ROS and matrix metalloproteinase-9 (MMP-9) activation. Nanoceria downregulate inducible nitric oxide synthase (iNOS) to decrease NO and ONOO^−^ levels. C60 can be localized in mitochondria to reduce lactate dehydrogenase (LDH) release. Liposome and polymeric nanoparticles can deliver antioxidants, antioxidant enzymes, and genes to minimize the free radicals. Most antioxidants activate nuclear factor-erythroid 2-related factor 2 (Nrf2), advance Nrf2 translocation to the nucleus, and bind with antioxidant response element (ARE) to promote the expression of ARE-derived antioxidant gene. Reprinted from [27] under the terms of the Creative Commons Attribution License.

**Figure 11 nanomaterials-12-03397-f011:**
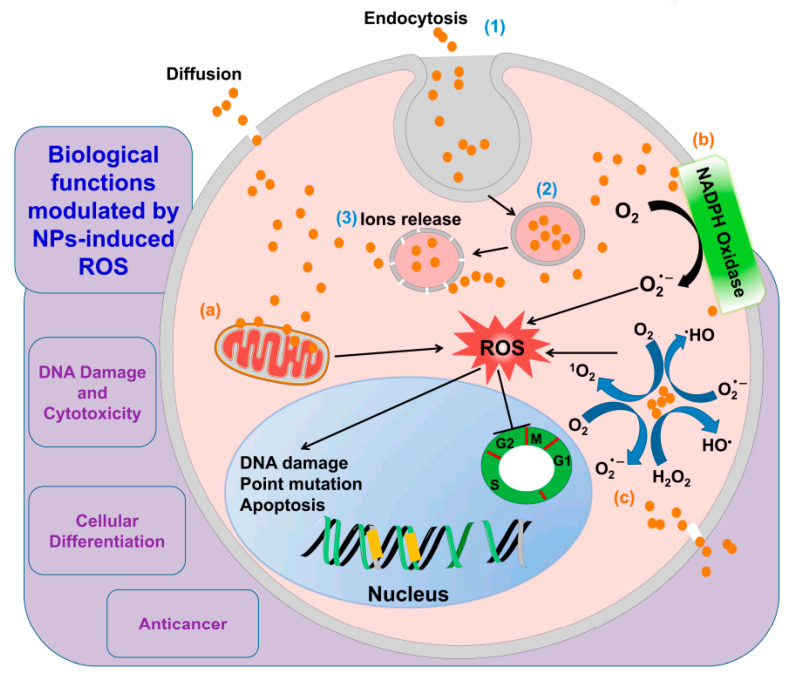
ROS generation mechanisms of NPs, after entering a cell via a sequence of (1) endocytosis; (2) endocytotic vesicle formation; and (3) ion release from the vesicles into the intracellular medium. ROS is generated by the NPs through (**a**) interaction with the mitochondria; (**b**) interaction with NADPH oxidase; and (**c**) factors related to the physicochemical properties, such as size, shape, photoreactive properties, and surface chemistry. Reprinted from [316] under the terms of the Creative Common CC BY license.

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
