# Peer review of "Possible Synergies of Nanomaterial-Assisted Tissue Regeneration in Plasma Medicine: Mechanisms and Safety Concerns"

_nanomaterials, 2022, doi:10.3390/nano12193397_

Round 1

Reviewer 1 Report

This is a comprehensive review, which summarize the application of nanomaterials and plasma in regenerative medicine and tissue engineering, also summarize synergies between plasma and nanomedicine, the safety concerns. Following issue need to be addressed well before publication acceptance.

1.     P5, line 165, what size and composition should the NPs have to activate cells through oxidative stress-mediated pathways?

Line 167, ROS are known to promote what kind of disease?

The description from line 167 to line 170 is too vague, what’s “certain level” refers to? how much ROS could cause severe biological dysfunction and has effect on cell survival?

2.     Line 762, low penetration depth in tumor?

3.     Line 763, what’s straightforward manner refers to?

4.     Please go through the whole manuscript and make sure you have full names of all abbreviations.

5.     Line 787, what’s “better distribution” refers to? please clarify.

6.     Please include more description about Fig 8.

Reviewer 2 Report

Cold atmospheric plasma and nanomedicine originally emerged as individual domains, but are increasingly applied in combination with each other during recent years. In this perspective article, the authors therefore start from the fundamental mechanisms in the individual technologies, in order to envision possible synergies, as well as research strategies to discover and optimize them. They emphasize the toxicity concerns in plasma and nanomedicine, which may be partly remediated by their combination, but also partly amplified. A widespread use of standardized protocols and materials is therefore strongly recommended, to ensure both a fast and safe clinical implementation.

The author has conducted extensive literature research, the following changes are required before the manuscript can be accepted:

General Comments.

1.        Keywords need to be carefully considered and appropriately reduced.

2.        The language expression of the manuscript is too complicated. It should be properly embellished to make it more concise.

3.        Please confirm whether the figures in the manuscripts are original by the author? If not, take care to cite relevant references.

4.        Explain all the acronyms in this manuscript when they first appeared.

5.        On the basis of reasonable citations, references should be abridged.

6.        Please check all the writing mistakes in the manuscript.

Reviewer 3 Report

The work entitled “Possible synergies of nanomaterial-assisted tissue regeneration in plasma medicine: mechanisms and safety concerns” reviews the possible synergies between nanomaterials and cold plasma, in which way they enhance each other and overcome each other limitations. The authors also report on the potential toxicity of this process and the biological effects implied. Considering this research is directed to tissue regeneration, the authors should better emphasize that in the abstract. The work is very well written and organized. The authors were clear in delineating the most important areas of research to explore in this review. The work is of importance to the field, focuses on recent research (without overlooking old concepts) and is scientifically sound. I recommend its publication after minor revision, addressing the details listed below:

-          I would recommend the elimination of the last paragraph of the introduction as it is not necessary.

-          In figure 1, it would be more interesting to see as well possible outcomes of using one strategy and the other being represented. Figure 2 quality must be improved.

-          Copyrights permissions from the publications should be acquired for some images.
